# Glycolytic reliance promotes anabolism in photoreceptors

Yashodhan Chinchore[1], Tedi Begaj[1], David Wu[1], Eugene Drokhlyansky[1], Constance L Cepko[1,2]*

[1]Departments of Genetics and Ophthalmology, Harvard Medical School, Boston, United States; [2]Howard Hughes Medical Institute, Harvard Medical School, Boston, United States

**Abstract** Vertebrate photoreceptors are among the most metabolically active cells, exhibiting a high rate of ATP consumption. This is coupled with a high anabolic demand, necessitated by the diurnal turnover of a specialized membrane-rich organelle, the outer segment, which is the primary site of phototransduction. How photoreceptors balance their catabolic and anabolic demands is poorly understood. Here, we show that rod photoreceptors in mice rely on glycolysis for their outer segment biogenesis. Genetic perturbations targeting allostery or key regulatory nodes in the glycolytic pathway impacted the size of the outer segments. Fibroblast growth factor signaling was found to regulate glycolysis, with antagonism of this pathway resulting in anabolic deficits. These data demonstrate the cell autonomous role of the glycolytic pathway in outer segment maintenance and provide evidence that aerobic glycolysis is part of a metabolic program that supports the biosynthetic needs of a normal neuronal cell type.

## Introduction

Sensory neurons capture information from the environment and convert it into signals that can greatly impact the survival of an organism. These systems are thus under heavy selective pressure, including for the most efficient use of energy to support their sensitivity and efficiency (*Niven and Laughlin, 2008*). In this regard, the primary photoreceptor cells face a dual challenge. They not only need to preserve their membrane excitability via ion pumps by ATP hydrolysis (*Okawa et al., 2008*) but also maintain elaborate membranous organelles (rhabdomeres in invertebrates and outer segments in vertebrates) that maximize light capture. The maintenance of such structures requires considerable metabolic resources. Invertebrate photoreceptors exhibit light-dependent endocytosis of their photosensitive membranes (*Satoh et al., 2005*) enabling the recycling of these resources. In contrast, vertebrate photoreceptors shed a fraction of their outer segments (OS) daily, to be phagocytosed by the juxtaposed retinal pigmented epithelium (RPE) (*Basinger et al., 1976*; *LaVail, 1976*) (*Figure 1—figure supplement 1*). To sustain a constant volume of the OS, a cell must channel metabolites toward biosynthesis, against the backdrop of very high ATP consumption, which is required to maintain membrane potential. Photoreceptors thus must balance the use of their intracellular carbon pool between oxidative catabolism, to generate the required ATP, and anabolism, to continually renew the OS.

The mammalian retina depends upon glucose and glycolysis for survival and function (*Chertov et al., 2011*; *Noell, 1951*). The majority (~80%) of glucose is converted to lactate via glycolysis (*Cohen and Noell, 1960*; *Warburg, 1925*; *Winkler, 1981*). The adult retina can produce lactate aerobically (aerobic glycolysis/Warburg effect) with an ~50% increase during anaerobic conditions (Pasteur effect) (*Cohen and Noell, 1960*). The cell types that carry out aerobic glycolysis in the normal adult retina have not been determined. The photoreceptors have been assumed to

*For correspondence: cepko@
genetics.med.harvard.edu

**Competing interests:** The authors declare that no competing interests exist.

**eLife digest** Living cells need building materials and energy to grow and carry out their activities. Most cells in the body use sugars like glucose for these purposes. In a process known as glycolysis, cells break down glucose into molecules that are eventually converted to carbon dioxide and water to form the chemical ATP – the cellular currency for energy. Developing cells that have not yet fully specialized, and rapidly dividing cells, like cancer cells, consume large amounts of glucose via aerobic glycolysis (also known as the Warburg effect) as they require high levels of energy and building materials. As cells become more specialized and divide less often, they have a reduced need for building blocks, and adjust their consumption and breakdown of glucose accordingly.

One exception is the photoreceptor cells, found in the light-sensitive part of our eyes. Although these specialized cells do not divide, they still need a lot of energy and building blocks to constantly renew their light-sensing and processing structures, and to capture and convert the information from the environment into signals. Previous research has shown that the eye also uses the Warburg effect. However, until now, it was not known whether the photoreceptors or other cells in the eye carry out this form of glycolysis.

Using genetic tools, Chinchore et al. analysed how the photoreceptor cells in mice used glucose. The experiments demonstrated that the photoreceptors do indeed carry out the Warburg effect. Chinchore et al. further discovered that the Warburg effect is regulated by the same key enzymes and signalling molecules that cancer cells use. This indicates that specialized cells like photoreceptors might choose to retain certain metabolic features of their precursor cells, if they need to.

These findings provide new insight into how photoreceptors use glucose. The next step will be to understand how aerobic glycolysis is regulated in healthy eyes as well as in eyes that are affected by degenerating diseases, which may ultimately lead to new ways of treating blindness.

rely on aerobic glycolysis. This assumption is based on the adverse effects on photoreceptor function after *en masse* inhibition of whole retinal glycolysis by pharmacological treatments e.g. with iodoacetate (*Winkler, 1981*). The Warburg effect exemplifies an elaborate set of metabolic strategies adopted by a cell to preferentially promote glycolysis (*Gatenby and Gillies, 2004*; *Liberti and Locasale, 2016*). One drawback of inhibiting glycolytic enzyme activity in the retina is that such a manipulation does not differentiate between aerobic glycolysis and housekeeping glycolysis- a pathway critical for most cell types.

Studies conducted on retinal tissue *in vitro* indicate that isolated mammalian photoreceptors can consume lactate, which can be produced by glycolysis in retinal Mueller glia (*Poitry-Yamate et al., 1995*). Thus, the decreased photoreceptor function after whole retinal glycolytic enzyme inhibition could be a non-cell-autonomous effect on Muller glia. Although many features of the 'lactate shuttle' and its *in vivo* relevance have recently been questioned (*Hurley et al., 2015*), it is important to devise an experimental strategy that would be able to discern the cell-autonomous versus non-autonomous requirement of glycolysis for the photoreceptors.

The cellular origins and purpose of aerobic glycolysis in the retina, its relevance to photoreceptor physiology, and its regulation, are not understood. In this study, we explored the propensity of photoreceptors to produce or consume lactate and utilized genetic manipulations to reveal the regulatory mechanisms of glycolysis. We show that rod photoreceptors rely on glycolysis for their OS biogenesis. Genetic perturbations targeting allostery or key regulatory nodes in the glycolytic pathway impacted the OS size. Fibroblast growth factor (FGF) signaling was found to regulate glycolysis, with antagonism of this pathway resulting in anabolic deficits. These data demonstrate the cell autonomous role of the glycolytic pathway in OS maintenance and provide evidence that aerobic glycolysis is part of a metabolic program that supports the biosynthetic needs of a normal neuronal cell type.

## Results

### Aerobic glycolysis in the retina

We first examined lactate production from the retina and assayed the metabolic consequences of inhibiting aerobic glycolysis. Lactate is produced by reduction of pyruvate, a reaction catalyzed by lactate dehydrogenase (LDH) (*Figure 1—figure supplement 2A*). Freshly isolated retinae were cultured in the presence or absence of sodium oxamate- an LDH inhibitor. These were subsequently transferred to buffered Krebs'-Ringer's medium that has glucose as the sole source of carbon (see - an LDH inhibitor. These were subsequently transferred to buffered Krebs'-Ringer's medium that has glucose as the sole source of carbon (see Materials and methods), and lactate secretion was quantified (*Figure 1A*). The extracellular secreted lactate was measured because it represents the pyruvate-derived carbons that are diverted away from other intracellular metabolic processes or the mitochondria. Oxamate treatment led to a significant drop in the secreted lactate production rate compared to control. In addition, the ATP levels were monitored and, surprisingly, the steady-state levels of ATP in oxamate-treated retinae did not differ from the control retinae (*Figure 1B*). This could be due to a relatively minor glycolytic contribution to the total ATP pool, a compensatory metabolic realignment toward mitochondria-dependent ATP production or existence of phosphotransfer enzyme systems such as adenylate kinase or creatine kinase. To differentiate among these possibilities, explants were cultured in oxamate or control conditions followed by a short treatment with $NaN_3$ to inhibit mitochondrial ATP synthesis (*Figure 1B*). Control retinae displayed ~50% reduction in ATP levels after incubation in $NaN_3$. Interestingly, oxamate-treated retinae displayed a further 20% decrease in ATP after exposure to $NaN_3$. Thus, inhibiting lactate synthesis resulted in a greater fraction of the ATP pool that was sensitive to mitochondrial function.

### Lactate producing isoform of LDH in photoreceptors

Next, we wanted to ascertain if photoreceptors produce or consume lactate. As a first step, the expression of the LDH subtypes was examined. LDH is a tetrameric enzyme composed of LDHA and LDHB subunits encoded by the *Ldha* and *Ldhb* genes respectively. The subunits can assemble in five different combinations with differing kinetic properties (*Dawson et al., 1964*; *Doherty and Cleveland, 2013*) (*Figure 1—figure supplement 2B*). A tetramer of all LDHA subunits has high affinity for pyruvate and a higher $V_{max}$ for pyruvate conversion to lactate than does an all-LDHB isoenzyme. In addition, many glycolytic cancers have elevated *Ldha* expression (*Balinsky et al., 1983*; *Behringer et al., 2003*). On the contrary, an all-LDHB tetramer is maximally active at low pyruvate concentrations, is strongly inhibited by pyruvate, and is expressed in tissues using lactate for oxidative metabolism or gluconeogenesis (*Dawson et al., 1964*). We examined the expression of LDHA and LDHB subunits in the retina by immunohistochemistry (IHC) (*Figure 1C*). Photoreceptors showed strong expression of LDHA, particularly with respect to the other retinal cell types. Similar results were obtained by others using a different set of commercially available antibodies (*Casson et al., 2016*; *Rueda et al., 2016*). Immunohistochemical localization also indicated that LDHB was abundantly expressed in the cells of the inner nuclear layer (INL), which includes interneurons and Mueller glia. To validate the staining pattern obtained by IHC, expression analysis of transcripts of *Ldha* and *Ldhb* genes by in situ hybridization (ISH) was performed (*Figure 1D*). This confirmed that *Ldha* expression is enriched in the photoreceptors, whereas *Ldhb* is excluded. This was also confirmed by qRT-PCR analysis of *Ldha* and *Ldhb* transcripts in isolated rod photoreceptor cDNA (*Supplementary file 1*). We conclude that photoreceptors have predominantly LDHA-type subunits.

We also assessed the ability of the retina to secrete lactate after treatment with the LDHA-specific inhibitor, FX-11 (*Le et al., 2010*). FX-11 significantly reduced lactate secretion (*Figure 1E*). Similar to oxamate, FX-11 also resulted in an increased percentage of ATP that was sensitive to azide inhibition (*Figure 1F*). To investigate if photoreceptors produce lactate in an *Ldha*-dependent manner, mice with a conditional allele of *Ldha* (*Wang et al., 2014b*), (*Ldha*^fl/fl^), were used. The specificity and efficiency of Cre recombinase under control of the rhodopsin regulatory elements (*Le et al., 2006*) were first tested, which showed that only rod photoreceptors had a history of *cre* expression (*Figure 1—figure supplement 2C,D*) (the mouse line henceforth called Rod-cre). The recombination efficiency varied between ~50–90% of photoreceptors among different retinae. The Rod-cre; *Ldha*^fl/fl^ retinae were examined for LDHA protein, which showed a significant reduction (*Figure 1—figure*

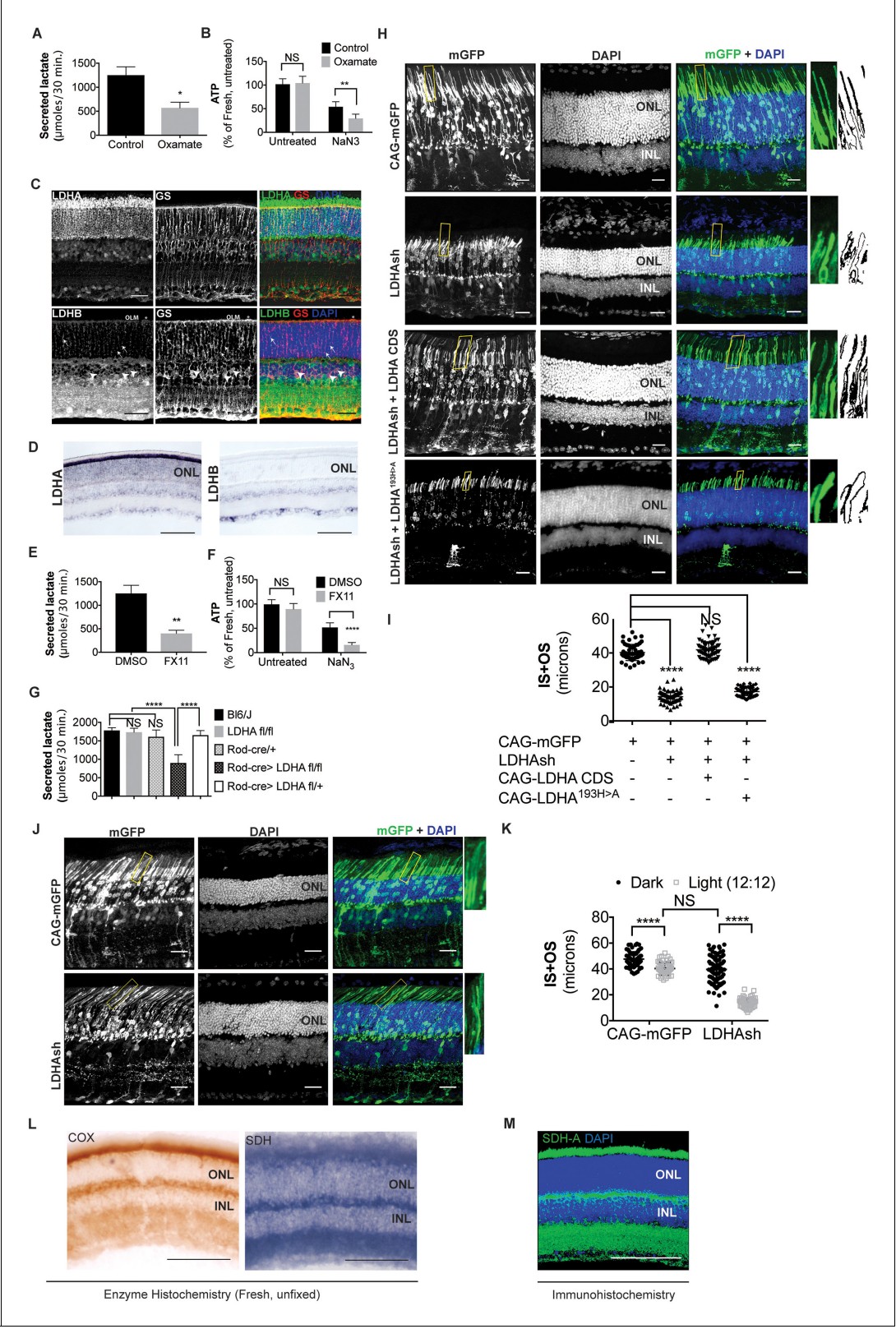

**Figure 1.** *Ldha*-dependent aerobic glycolysis and outer segment maintenance in photoreceptors. (**A**) Freshly explanted retinas were treated with the LDH inhibitor, sodium oxamate, for 8 hr in explant culture medium, transferred to Krebs'-Ringer's for 30 min, and lactate was measured in the supernatant. Control (n = 5), Oxamate (n = 4). (**B**) Freshly explanted retinas were treated with oxamate or NaCl (control) in explant culture medium for 8 hr, followed by treatment with $NaN_3$ or NaCl (untreated group) in Krebs'-Ringer's medium for 30 min. ATP per retina was then measured. *n* = 7, Control

*Figure 1 continued on next page*

*Figure 1 continued*

untreated; n = 8, Oxamate untreated n = 8, Control NaN$_3$; n = 8, Oxamate NaN$_3$. (**C**) Expression of *Ldha* and *Ldhb* as determined by IHC. Glutamine synthetase (GS), a Mueller glia-specific marker, colocalized with LDHB in the cell bodies (arrowheads), processes ensheathing the photoreceptors (arrows) and the outer limiting membrane (OLM, *). Scale bar, 50 µm. (**D**) ISH for *Ldha* and *Ldhb*. *Ldha* RNA displayed photoreceptor-enriched expression while *Ldhb* RNA was not observed in photoreceptors. Scale bar, 100 µm. (**E, F**) Freshly explanted retinas were treated with FX11 or DMSO for 8 hr and transferred to Krebs'-Ringer's for 30 min and secreted lactate was measured (**E**) n = 5, DMSO; n = 6, FX11, or they were transferred to Krebs'-Ringer's buffer with NaN$_3$ or NaCl (untreated group) for 30 min for ATP quantitation (**F**). ATP per retina was measured at the end of the assay. n = 8, DMSO untreated; n = 8, FX11 untreated; n = 9, DMSO NaN$_3$; n = 7, FX11 NaN$_3$. (**G**) Freshly explanted retinae were transferred to Krebs'-Ringer's for 30 min and secreted lactate was measured. n = 8, Bl6/J; n = 8, *Ldha*$^{fl/fl}$; n = 8, Rod-cre; n = 16, Rod-cre> *Ldha*$^{fl/fl}$; n = 8, Rod-cre> *Ldha*$^{fl/+}$. (**H**) Photoreceptor outer segment phenotype 42–45 days following in vivo electroporation of a knock-down construct (shRNA) for Ldha. CAG-mGFP was used for coelectroporation. Plasmid combinations listed on the left. Magnification of areas outlined in yellow is displayed on right with threshold-adjusted rendering to highlight inner and outer segments. Scale bar, 25 µm. (**I**) Quantification of inner+outer segment (IS+OS) lengths. n = 53–74 photoreceptors, 4–5 retinae. (**J**) Photoreceptor outer segment phenotype of dark-reared animals. Electroporated pups were transferred to dark on the day of eye opening (P11) and reared with their mothers for 3 weeks. (**K**) Quantification of inner+outer segment lengths of (**J**). n = 53–83 photoreceptors, 4–5 retinae. (**L**) Colored end products of redox reactions catalyzed by COX and SDH enzymes in retinal tissue. Scale bar, 200 µm. (**M**) IHC for SDH-A subunit in adult retina. Scale bar, 200 µm. ONL, outer nuclear layer. INL, inner nuclear layer. Data, Mean±SD. Statistics, unpaired, two-tailed *t*-test with Kolmogorov-Smirnov correction for panels **A**, **E**; two-way ANOVA with Tukey's correction for panels **B**, **F** and **K**; one-way ANOVA with Tukey's multiple comparison test for panels **G**, **I**.

The following source data and figure supplements are available for figure 1:

**Source data 1.** Source data for *Figure 1A,B,E,F and G*.

**Figure supplement 1.** Metabolic challenges of photoreceptor cell.

**Figure supplement 2.** Characterization of *Ldha* knockdown and mitochondrial function.

**Figure supplement 3.** Cell-autonomous effect of *Ldha* knockdown.

**Figure supplement 4.** Mitochondrial activity after *Ldha* loss-of-function.

---

*supplement 2E*). A compensatory expression of *Ldhb* in photoreceptors was not detected (*Figure 1—figure supplement 2F*). Lactate production in these retinae was examined and was found to be significantly reduced (*Figure 1G*). Thus photoreceptors produce lactate in an *Ldha*-dependent manner.

## Active LDHA supports outer segment biogenesis

In order to assess if reduction in *Ldha* expression created a cellular phenotype in photoreceptors, and if so, whether it was required autonomously, electroporation of a short hairpin RNA (shRNA) specifically targeting the 3' untranslated region (UTR) of the *Ldha* transcript was used (*Figure 1—figure supplement 2G*). This strategy was taken, vs.examination of the rods in the Rod-cre; *Ldha*$^{fl/fl}$ retinae, due to the concern that a reduction in lactate by rods might affect closely associated cell types, such as Mueller glia and/or RPE cells, creating non-autonomous effects on rods. A plasmid encoding this shRNA was delivered to the retina in vivo by electroporation. Electroporation occurs in patches comprising 15–30% of the retina, and in a given patch, only ~20% cells are electroporated (Sui Wang and C. Cepko, unpublished). Thus, plasmid transfection via electroporation allowed us to determine if *Ldha* has a cell-autonomous role in photoreceptors. The electroporated photoreceptors had markedly reduced OS length when compared to control (*Figure 1H,I*). Genetic complementation by coelectroporation of a sh-resistant *Ldha* cDNA that lacks the 3'UTR demonstrated that the defect was attributable to *Ldha* loss-of-function (*Figure 1H,I*) and the phenotype observed with the shRNA was not due to off-target effects. To determine if the catalytic activity of LDHA was required for rescue, an allele of *Ldha* with a point mutation in the catalytic center (LDHA$^{H193>A}$) was introduced. It failed to rescue the shRNA phenotype. Finally, expression of *Ldhb* was not sufficient to compensate for *Ldha* loss-of-function (*Figure 1—figure supplement 2H*). To confirm that the *Ldha* knockdown via electroporation conferred a cell-autonomous phenotype, we examined rhodopsin localization in mGFP-negative rods within the electroporated patch (*Figure 1—figure supplement*

*3*). A non-autonomous deleterious effect on rods that did not receive the plasmid (mGFP⁻) was not observed. The rhodopsin localization and the length of the OSs in GFP-negative rods within the patch did not vary from that of the rods lying outside of the electroporated patch (*Figure 1—figure supplement 3*).

The cyclical process of OS shedding and renewal is regulated by light (*LaVail, 1976*). Since LDHA function is necessary to maintain OS length, we assessed the effect of *Ldha* knockdown in dark-reared mice and compared with mice raised in normal room light. Electroporated mice were raised with their mothers in normal room light until eyes were open (P11), and then shifted to the dark for 3 weeks. In mice with no *Ldha* knockdown, there was ~25% increase in IS+OS length after dark rearing compared to the light:dark condition (*Figure 1J,K*), presumably as a part of an adaptive mechanism that might include less OS shedding (*Penn and Williams, 1986*). Interestingly, in mice with the *Ldha* knockdown, dark rearing resulted in a partial rescue of the *Ldha* knockdown phenotype (*Figure 1J,K*). The average IS+OS length after *Ldha* knockdown was similar to that of light-treated control animals. These data indicate that reducing the need for OS biogenesis, as occurs in the dark, led to a reduced reliance on *Ldha* function.

## Functional mitochondria in photoreceptors

Cells with immature or dysfunctional mitochondria become reliant on glycolysis by increasing *Ldha* expression at the expense of *Ldhb* (*Facucho-Oliveira et al., 2007*; *Ross et al., 2010*; *Trifunovic et al., 2004*). Although photoreceptors have abundant mitochondria, a reason for their high *Ldha* and low *Ldhb* expression could be subpar mitochondrial function, especially when compared to other retinal cell types. Thus, we assessed whether there was a mitochondrial activity difference between the photoreceptors and INL cells by examining succinate dehydrogenase (SDH) and cytochrome oxidase (COX) activity in fresh, unfixed, adult retinal sections (*Figure 1L*). SDH/complex II plays a role in the citric acid cycle, as well as in the electron transport chain, and its subunits are encoded by the nucleus. COX or complex IV plays a role in the electron transport chain and has catalytic subunits that are encoded by the mitochondrial genome (mtDNA). SDH activity was not lower in the photoreceptors relative to INL cells. COX activity was high in the photoreceptor layer, even higher than that seen in the other retinal layers. The specificity controls for the histochemical reaction are presented in *Figure 1—figure supplement 2I*. Finally, IHC for SDH was carried out. The highest IHC signal was observed in the photoreceptor inner segments (IS), as well as the OPL and IPL synaptic layers (*Figure 1M*), in good agreement with the observed SDH activity. IHC for another mitochondria-specific enzyme, pyruvate dehydrogenase, showed a similar pattern (*Figure 1—figure supplement 2J*) indicating that these are the sites of maximal mitochondrial densities in the retina. These data align with other studies that assessed mitochondrial activity in the retina (*Hansson, 1970*; *Rueda et al., 2016*). Thus, lactate production by the photoreceptors cannot be attributed to lack of mitochondrial activity. Similarly, a decrease in mitochondrial COX activity was not detectable in the Rod-cre; *Ldha^{fl/fl}* retinae (*Figure 1—figure supplement 4*) by the histochemical assay.

## Allosteric regulation of glycolysis in photoreceptors

LDHA supports glycolysis by providing a ready supply of cytosolic NAD⁺ that is independent of O₂ availability and/or mitochondrial function. The phenotype observed following *Ldha* knockdown might be indicative of a reliance on glycolysis where cells might exhibit a preference for unabated and rapid flux through glycolysis. Alternatively, it could be due to an unidentified role of *Ldha* in OS maintenance. To understand the extent of photoreceptors' dependence on glycolysis, we designed an experimental strategy that satisfied the following criteria: (1) Does not ablate core glycolytic enzymes in order to avoid pleiotropic effects due to their possible non-glycolytic roles, (2) Targets a glycolytic node such that impact on other biosynthetic pathways, such as Pentose Phosphate Pathway (PPP), would be minimal and (3) Uncovers glycolytic reliance and differentiates it from 'housekeeping' glycolysis. Glucose-derived metabolites are committed towards glycolytic flux by the enzyme 6-phosphofructo-1-kinase (PFK1), which catalyzes conversion of fructose-6-phosphate (F6P) to fructose-1,6-bisphosphate (F-1,6-BP) (*Figure 2—figure supplement 1.*). The most potent allosteric activator of PFK1 is fructose-2,6-bisphosphate (F-2,6-BP) (*Hers and Van Schaftingen, 1982*). F-2,6-BP is synthesized from F6P by the kinase activity of the bifunctional enzyme, 6-phosphofructo-2-kinase/fructose-2,6-bisphosphatase (PFK2) (*Figure 2—figure supplement 1A,B*). To examine the

glycolytic dependence of photoreceptors, we targeted the steady-state levels of the metabolite, F-2,6-BP as it would satisfy the above criteria.

First, we examined expression of PFK2 isoenzymes encoded by *Pfkfb1-4* genes (*Figure 2—figure supplement 1C*). *Pfkfb3* expression could not be detected. *Pfkfb1*, *2* and *4* were expressed in either a photoreceptor-enriched or photoreceptor-specific pattern, suggesting a propensity of these cell types to regulate glycolysis via a PFK2-dependent mechanism.

With the exception of Pfkfb3, all other PFK2 isoenzymes have kinase and phosphatase domains on the same polypeptide (*Mor et al., 2011*) (*Figure 2—figure supplement 1B*). In addition to potential problems posed by functional redundancy (i.e. knockdown of one isoenzyme might not be sufficient), genetically ablating the PFK2 isoforms would not uncover the preference for directionality (i.e. an observed phenotype could be attributed to absence of either the kinase or phosphatase function). In addition, the structure-function relationships of their kinase and phosphatase domains are not known, thus making kinase- or phosphatase-dead versions is not straightforward. To overcome these problems, we overexpressed *Tigar* (*TP53-induced glycolysis and apoptosis regulator*) as it is functionally similar to the phosphatase domain of PFK2 with well-characterized F-2,6-BPase activity (*Bensaad et al., 2006*) (*Figure 2—figure supplement 1B*) and hence reduces the steady state levels F-2,6-BP. This is the intended effect and bypasses the aforementioned concerns with the conventional genetic loss-of-function approach associated with PFK2 isoenzymes. In addition to the predicted function of reducing glycolysis, overexpression of *Tigar* would not negatively affect the PPP (*Bensaad et al., 2006*).

We utilized an experimental strategy that addressed the following concerns: (1) The effect should be autonomous to photoreceptors, (2) the phenotype should be induced in fully mature photoreceptors, and (3) the phenotype should discernably be due to perturbations specifically of fructose-2,6-bisphosphate. Our experimental scheme utilized a construct that expressed tamoxifen-inducible Cre only in rods (*Figure 2A,B* and *Figure 2—figure supplement 1D*). Expression of *Tigar* specifically in adult photoreceptors resulted in a significant reduction of OS length (*Figure 2C,D*). This phenotype was specifically attributable to the phosphatase activity because expression of a catalytic dead version of Tigar, Tigar-TM (triple mutant, H11>A, E102>A, H198>A)(*Bensaad et al., 2006*), did not cause a change in the photoreceptor OS length (*Figure 2—figure supplement 1E,F*). To ascertain if the phenotype is specifically attributable to Tigar's phosphatase activity on F-2,6-BP, we decided to coexpress *Pfkfb3*- a PFK2 isoform that has the kinase activity ~700 fold higher than the phosphatase (*Sakakibara et al., 1997*) (*Figure 2A* and *Figure 2—figure supplement 1B*). Interestingly, overexpression of *Pfkfb3* alone did not result in an overt phenotype- the OS length and morphology were indistinguishable from those of the control electroporated retina (*Figure 2C,D*). Overexpression of *Pfkfb3* was able to rescue the reduction in OS length caused by *Tigar* expression (*Figure 2C,D*). Together, these data suggest that adult photoreceptors are sensitive to perturbations targeting F-2,6-BP.

Next, the effects of *Tigar* expression on glycolysis were assayed. Although electroporation answers the question of the cell autonomous effect of a perturbation, the total number of affected cells and their percentage in the electroporated area are too minor to determine biochemical contributions. Adeno-associated virus (AAV)-mediated transduction of Tigar into photoreceptors was thus used, as it transduces a greater percentage of cells than electroporation. An AAV construct that drives expression of *Tigar* and/or mGFP from the bovine rhodopsin (*RHO*) promoter specifically in rods was constructed (*Figure 2E,F* and *Figure 2—figure supplement 1G*). The AAVs (expressing mGFP alone or mGFP and TIGAR) were injected at postnatal day 6 (P6), after the end of cell proliferation, and nearly full retinal infection was confirmed by indirect ophthalmoscopy at P24-P27 by assessing the GFP fluorescence. Retinae were harvested at P28 and examined for expression (*Figure 2—figure supplement 1H*) and lactate levels (*Figure 2G*). Consistent with the idea that Tigar would interfere in allosteric regulation of glycolysis, a significant reduction in retinal lactate was observed in the AAV-TIGAR infected retinae compared to the control AAV-mGFP infected retinae (*Figure 2G*).

## Nonequivalent roles of pyruvate kinase isoforms

Given the essential role of *Ldha* in postmitotic photoreceptors and proliferating cancer cells, other aspects of metabolism that have been discovered in cancer cells, such as the expression of pyruvate kinase isoforms, were investigated. Pyruvate kinase catalyzes the final irreversible reaction of

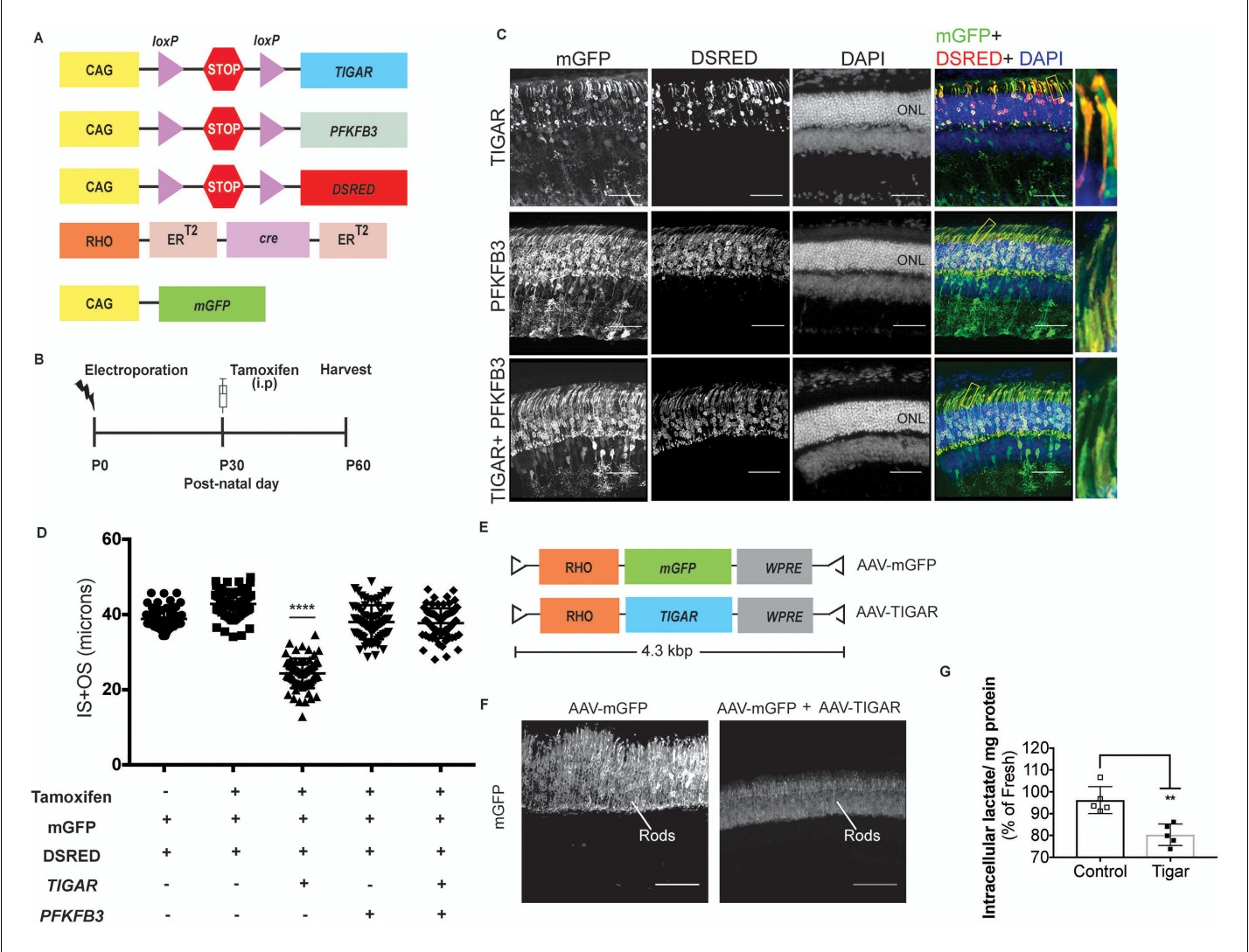

**Figure 2.** Targeting allostery reveals glycolytic reliance for outer segment maintenance. (A) Constructs for spatio-temporal control of expression of *Tigar* and *Pfkfb3*. DsRed used as the *cre* reporter, mGFP as a coelectroporation marker. (B) Scheme for electroporation and tamoxifen induction. i.p, Intraperitoneal. (C, D) IS+OS length were measured following introduction of *Tigar* (n = 72 cells), *PFKB3* (n = 72 cells), and *Tigar* and *Pfkfb3* constructs (n = 78), shown in (A). Controls were -tamoxifen (n = 62) and +tamoxifen (n = 74). Scale bar, 50 µm. Data are Mean±SD. One-way ANOVA with Tukey's correction. Outlined areas magnified to show IS and OS morphology. (E) AAV genomes for expression of mGFP (AAV-mGFP) or *Tigar* (AAV-TIGAR). (F) Cross-sections of AAV-mGFP alone or AAV-TIGAR (coinjected with AAV-mGFP) infected retinae harvested at P28 imaged for mGFP expression. (G) Intracellular lactate normalized for total protein was quantified for retinae infected with AAV-mGFP (Control) or AAV-mGFP + AAV-TIGAR. Data represented as percentages relative to age-matched, freshly isolated retinae. Data, Mean±SD. Unpaired, two-tailed *t*-test with Kolmogorov-Smirnov correction for non-Gaussian distribution. ONL, outer nuclear layer.

The following source data and figure supplement are available for figure 2:

**Source data 1.** Source data for *Figure 2G*.

**Figure supplement 1.** Allosteric regulation in photoreceptor glycolysis.

glycolysis and distinct isoenzymes are encoded by two genomic loci, *Pkm* (muscle) and *Pklr* (liver and red blood cell). *Pklr* transcripts were not detected in the retina, (*Figure 3—figure supplement 1*), but M1 and M2 splice isoforms of the *PKM* gene were detected (*Figure 3—figure supplement 2A*) in line with protein expression data reported earlier (*Lindsay et al., 2014*). The M2 isoform is

known to regulate aerobic glycolysis, promotes lactate production, and is upregulated in many tumors (*Christofk et al., 2008a*, *2008b*). This isoform was previously reported to be expressed in photoreceptors (*Casson et al., 2016*; *Lindsay et al., 2014*; *Morohoshi et al., 2012*; *Rajala et al., 2016*; *Rueda et al., 2016*). We confirmed that there is photoreceptor-enriched expression of PKM2 by IHC (*Figure 3A*). PKM1, known to be expressed in most differentiated cell types in adults (*Jurica et al., 1998*), was expressed in the cells of the INL and ganglion cell layer, as shown by IHC (*Figure 3A*), but was not detectable in photoreceptor cells. In this regard, our data differed from some published findings (*Casson et al., 2016*; *Lindsay et al., 2014*) but matched those of others (*Rajala et al., 2016*). To address this discrepancy and validate commercially available antibodies, we performed isoform-specific ISH (*Figure 3A*) and confirmed the expression pattern that we observed using IHC. We also examined transcript abundance by qPCR in mRNA purified from isolated rod photoreceptor cells (*Supplementary file 1*) and found the M1 isoform to be much less abundant than M2 in the photoreceptors.

Postnatally, PKM1 protein expression gradually increased, in correlation with increased differentiation and decreased proliferation in the developing retina (*Figure 3B*). On the other hand, PKM2 protein expression was detectable during the period of proliferation and its expression did not decrease with increased differentiation, likely due to retention of expression in differentiated photoreceptors. Previous studies on pyruvate kinase in the context of proliferation have suggested that loss-of-function of *Pkm2* reduces proliferation attributable to the glycolytic reliance of mitotic cells for growth (*Christofk et al., 2008b*; *Israelsen et al., 2013*). To assess if PKM2 plays an essential role in rod photoreceptors, an shRNA construct that specifically targeted mouse PKM2 (PKM2sh), but spared PKM*1* (*Figure 3—figure supplement 2B,C,D,E*), was generated. *In vivo* electroporation of a plasmid encoding PKM2-specific shRNA resulted in photoreceptors with significantly shorter OS than control (*Figure 3D,E*). This phenotype could be rescued by coelectroporation of a construct encoding human PKM2 cDNA (*Figure 3C,D*), which was not targetable by the shRNA (*Figure 3—figure supplement 2F*). Coelectroporation of plasmid encoding mouse PKM1 with PKM2sh did not rescue the OS length defect (*Figure 3C,D*). These data demonstrate that PKM1 and PKM*2* play nonequivalent roles in rod photoreceptors. In order to further investigate whether PKM2 was needed for an autonomous role in rods, we electroporated a low concentration of plasmid encoding the shRNA (*Figure 3—figure supplement 3*). In addition to few electroporated rods, there were very few electroporated INL cells. In this condition, electroporated rods displayed a similar OS phenotype to that observed with higher concentrations of PKM2sh (*Figure 3C*), that is, reduced OSs. We also generated an shRNA construct that targeted exon 4, which is shared between mouse PKM1 and PKM2 (PKM1 +2 sh) (*Figure 3—figure supplement 2C,D*). Electroporation of this construct resulted in a significant decrease in the OS length (*Figure 3—figure supplement 2G*). The photoreceptor morphology and OS length were the same as that observed following electroporation with PKM2sh. While complementation with human PKM2 was sufficient to rescue the IS+OS length defect, we noted some abnormalities with the morphology of some of the photoreceptor ISs and OSs (*Figure 3—figure supplement 2G*). In 4/6 retinae, many photoreceptors lacked clear borders of IS and OS, though in 2/6 retinae, the morphology closely resembled that of control retinae (*Figure 3—figure supplement 2G*).

The contribution of PKM2 to OS maintenance was further investigated in the retinae of dark-reared mice electroporated with PKM2sh (*Figure 3E*). Dark rearing significantly increased OS length in these animals (*Figure 3F*). Taken together, the results from dark-reared animals, in which *Ldha* or *Pkm2* was knocked down, indicate the requirement for the glycolytic pathway in OS maintenance. Since two different genes that promote aerobic glycolysis are necessary for the light-dependent maintenance of OS, the short OS phenotype is likely due to a reduced supply of the building blocks normally supplied by aerobic glycolysis.

In order to probe the biochemical effects of PKM2 reduction, lactate production was examined. Since electroporated retinae are not ideal for these experiments, mice that had a conditional deletion of *Pkm2* in rods were used. The *Pkm2*^fl/fl^ mouse strain, in which the M2-specific exon 10 was floxed (*Israelsen et al., 2013*), was crossed with the Rod-cre strain. The retinae with deficiency of PKM2 had a small but significant decrease in lactate production, as compared to the controls (*Figure 3G*). We also noted upregulation of PKM1 in these retinae (*Figure 3—figure supplement 2H*) similar to what has been reported before (*Israelsen et al., 2013*). However, as noted above, in rods electroporated with PHM2sh, PKM1 expression was not observed (*Figure 3—figure*

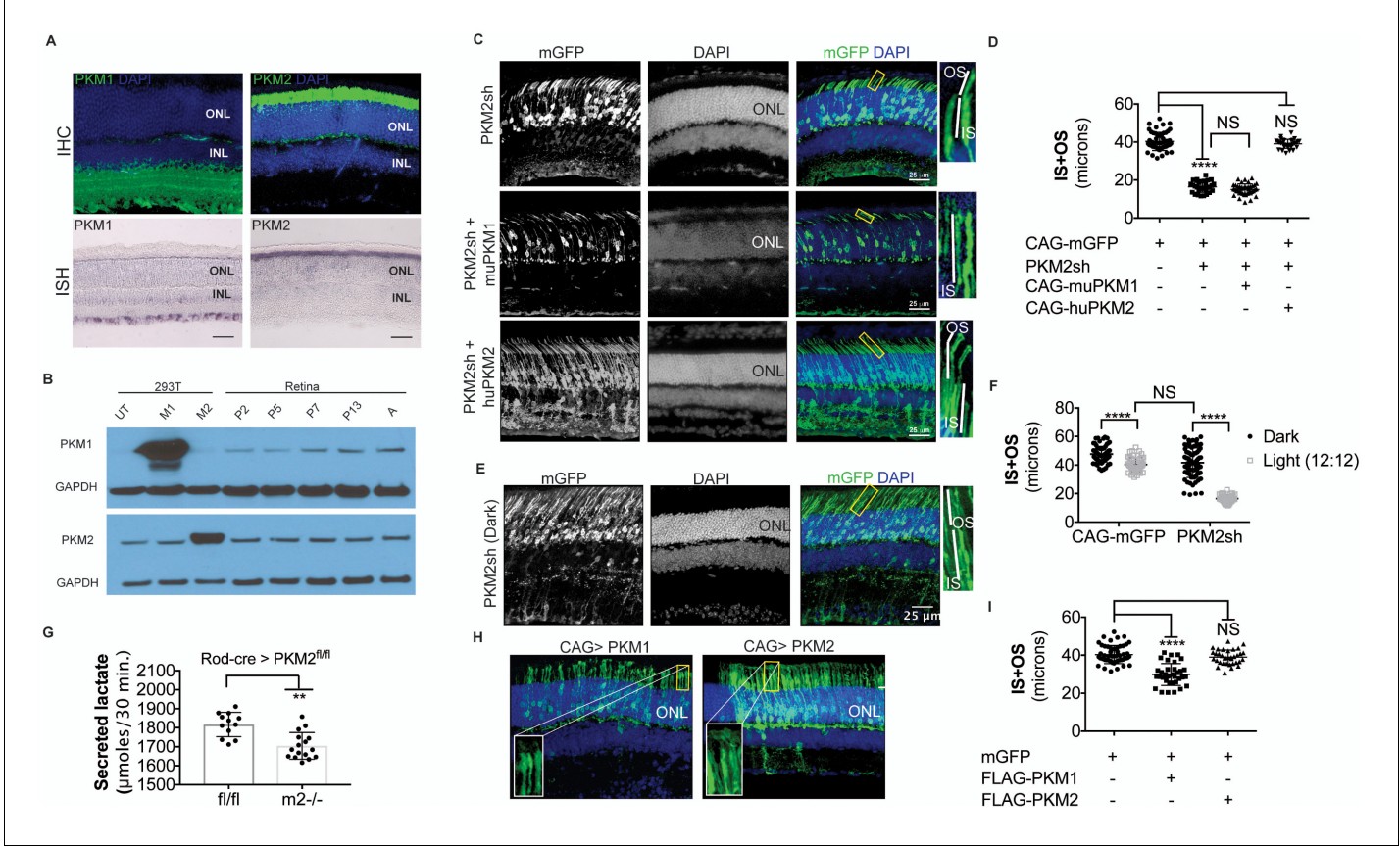

**Figure 3.** PKM1 and PKM2 isoforms have nonequivalent roles. (**A**) Biased expression of M1 and M2 isoforms in retinal layers detected by IHC and ISH. (**B**) Immunoblot of retinal lysates from postnatal retina at different developmental stages. HEK293T cell lysates that were from untransfected (UT) cells, or those transfected with CAG-FLAGmuPKM1 (M1) or CAG-FLAGmuPKM2 (M2) as controls. Postnatal age in days. A, mature retina (P25–P30). (**C**) Outer segment phenotype of P45 mice after electroporation with constructs encoding mouse PKM2-specific shRNA (PKM2sh) and adding either mouse PKM1 (muPKM1) or human PKM2 (huPKM2). Selected areas in yellow boxes are magnified on the right. (**D**) Quantification of IS+OS lengths obtained in (**C**). n = 32–53 cells from 3 to 4 retinae. (**E**) Outer segment phenotype of dark-reared P31 mice electroporated with PKM2sh-encoding plasmid. The yellow-boxed region is magnified and presented on the right. (**F**) Quantification of IS+OS lengths obtained in (e). n = 75 cells from three retinae. (**G**) Secreted lactate from freshly isolated retinae from *Pkm2*<sup>fl/fl</sup> (fl/fl) (n = 12) or Rod-cre> *Pkm2*<sup>fl/fl</sup> (m2<sup>−/−</sup>) (n = 16) mice. (**H**) Outer segment phenotype after CAG promoter-driven overexpression of Flag-tagged mouse PKM1 or PKM2. Inset, higher magnification of IS and OS. (**I**) Quantification of IS+OS lengths obtained in (**H**). n = 35 cells from three retinae in PKM1 and PKM2 groups. ONL, outer nuclear layer. Data, Mean±SD. Statistics, one-way ANOVA with Tukey's correction for panels **D**, **I**; two-way ANOVA with Tukey's multiple comparison test for panel **F**; unpaired, two-tailed *t*-test with Kolmogorov-Smirnov correction for panel **G**.

The following source data and figure supplements are available for figure 3:

**Source data 1.** Source data for *Figure 3G*.
**Figure supplement 1.** Assessment of *Pklr* expression in retina and liver.
**Figure supplement 2.** Characterization of PKM1 and PKM2 function in the retina.
**Figure supplement 3.** Cell-autonomous effect of PKM2 knockdown.
**Figure supplement 4.** PKM1 and PKM2 splicing factors.
**Figure supplement 5.** Outer segments in young Rod-cre; *Pkm2*<sup>fl/fl</sup> mice.
**Figure supplement 6.** Age-dependent retinal changes in Rod-cre; *Pkm2*<sup>fl/fl</sup> mice.

*supplement 2I*). One possibility for the difference in the presence of the M1-specific exon in the mRNA in the knockout vs. the knockdown manipulation might reflect a choice made by the splicing machinery. After the deletion of the 'preferred' M2-specific exon in the genome in the knockout, the splicing machinery might include the M1 exon as a default choice. However, when the shRNA was used to knockdown the PKM2 isoform, the splicing event that chose the M2-specific exon would have already happened.

The differential expression of the M1 and M2 isoforms in the retinal layers could be attributable to the differential expression of splicing factors that promote inclusion or exclusion of the M1- or M2-specific exon. To evaluate this possibility, we examined the expression of *Srsf3,* a splicing factor known to promote inclusion of the M2 exon (*Wang et al., 2012*), and *Ptbp1,* known to repress the M1 exon inclusion (*Chen et al., 2012*) (*Figure 3—figure supplement 4*). While *Srsf3* was expressed at higher levels in photoreceptors, *Ptbp1* was more enriched in the INL. Thus the regulation of *PKM* isoform preferences in retina is more complex than that predicted by canonical splicing models.

We also noted slightly reduced rod OS length in the region that had PKM1 expression in the young (postnatal 6 week) Rod-cre; *Pkm2^{fl/fl}* mice (*Figure 3—figure supplement 5*). Although the recombination in this line has been reported to be complete by 6 weeks, we cannot exclude the possibility of some recently recombined rods (that express PKM1) in this region. These rods might not have had enough time to have a discernable impact on rhodopsin abundance and OS length. In addition, there might be some non-recombined rods interspersed in the broad region where PKM1 expression was apparent, and contributed to longer OSs. Due to the packing density of rods and abundance of rhodopsin, an immunohistochemical approach might not be a suitable way to reliably assess the OS length and its relation to PKM2 function. We also examined older mice (37 weeks, ~8 months old) of this background hypothesizing that aging might uncover a subtle phenotype (*Figure 3—figure supplement 6*). The nuclei in the ONL region that had lost PKM2 protein expression were disorganized and lost their typical columnar arrangement, perhaps due to cell death. We also noted disorganization of OS in some regions where PKM2 was lost. Overall the OS length was reduced compared to aged Bl6/J mice, but this reduction was also noted for photoreceptors that had PKM2 protein expression. We cannot exclude the possibility of a non-autonomous effect on these rods as a response to tissue reorganization or alterations in their metabolic environment. Similarly, maintenance of some rhodopsin expression after PKM2 loss in the surviving rods might be an indication of an adaptive response on part of these cells. The electroporation approach, where only a few cells are transfected, circumvents these concerns and illustrates the critical cell-autonomous requirement of PKM2 for rods.

PKM1 is constitutively active while PKM2 is regulatable (*Anastasiou et al., 2012*). Biased expression of PKM2 in photoreceptors suggests that these cells may need to dynamically regulate glycolysis. The inability of PKM1 to rescue PKM2 loss-of-function indicates that merely replacing pyruvate kinase (PK) function after *Pkm2* knockdown is not sufficient to restore the OS. In addition, it indicates the importance of glycolytic regulation at the PK step in photoreceptors. We examined the effect of forced expression of PKM1 in the presence of endogenous PKM2, with the hypothesis that the constitutively active isoform might interfere at the regulatory step. We delivered plasmids encoding FLAG-tagged mouse PKM1 and PKM2 via *in vivo* electroporation (*Figure 3H*). Photoreceptors electroporated with PKM1-expressing constructs, but not PKM2 expressing constructs, had a reduction in the length of the OS (*Figure 3H,I*) with the majority of the photoreceptors in the PKM1 electroporated retinae lacking discernable OS. The two proteins were expressed at equivalent levels, as assessed by Western blotting for the FLAG epitope in HEK293T cells (*Figure 3—figure supplement 2J*).

## Fibroblast growth factor signaling regulates anabolism

PKM2 has been shown to interact with tyrosine phosphorylated proteins (*Christofk et al., 2008a*) and is tyrosine phosphorylated at position 105 (pY$^{105}$) in tumor cells (*Hitosugi et al., 2009*) leading to promotion of aerobic glycolysis. The pY$^{105}$ is a shared epitope in PKM1 and PKM2 (*Figure 4A*). To assess the phosphorylation status of PKM2 at this site, PKM2 was specifically immunoprecipitated from retinal lysates followed by immunoblotting using a phospho-Y$^{105}$-specific antibody (*Figure 4B*). We observed that PKM2 was phosphorylated at Y$^{105}$. In order to ascertain if phosphorylation of PKM2 at this site might have any physiological significance, its regulation by light was examined. PKM2 was immunoprecipitated from the retinae of mice at 3 hr intervals during a 24-hr time course,

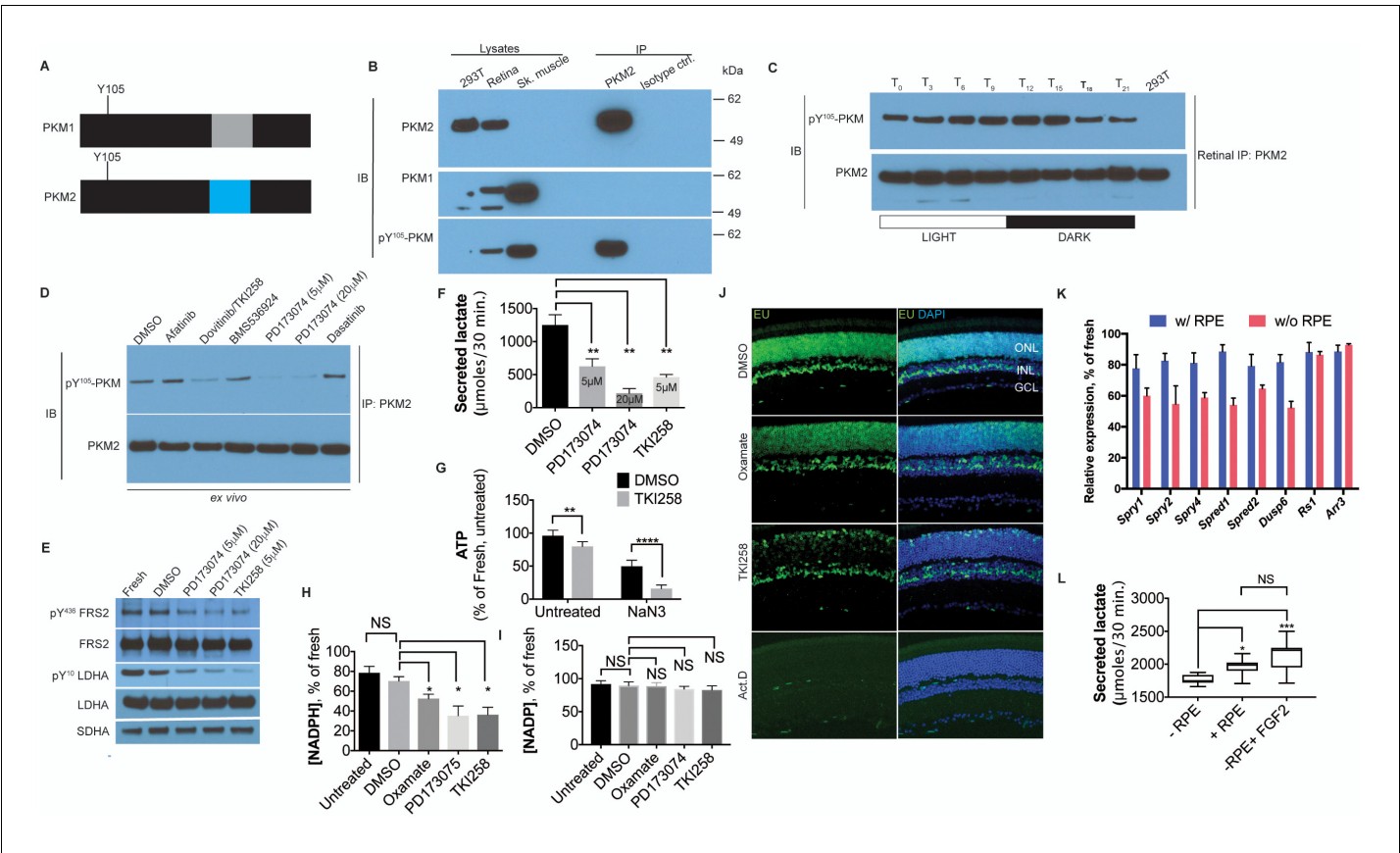

**Figure 4.** FGF signaling regulates aerobic glycolysis and anabolism. (A) Schematic of PKM1 and PKM2 polypeptide showing $Y^{105}$ is a shared epitope between PKM1 and PKM2. (B) Immunoprecipitation (IP) of PKM2 from adult retina followed by immunoblot (IB) for either PKM1, PKM2 or $pY^{105}$ PKM. IP using isotype-matched antibody (IgG) is used alongside to control for nonspecific binding. Lysates from skeletal muscle (expresses PKM1) and 293T (expresses only PKM2) included as controls. Molecular weight marker positions are depicted on the right-hand-side (C) Retinal lysates were prepared from eyes harvested at 3-hr interval during the 12 hr light 12 hr dark cycle. $T_0$ is the time point of light on in the room. The lysates were subjected to immunoprecipitation with anti-PKM2. Immunoprecipitates were probed for phosphorylation at $Y^{105}$ by immunoblotting with the phospho-specific antibody. (D) Lysates from explants treated with candidate tyrosine kinase pathway inhibitors or vehicle control (DMSO) were subjected to immunoprecipitation with anti-PKM2. Immunoprecipitates were probed for phosphorylation at $Y^{105}$ by immunoblotting with the phospho-specific antibody. (E) FGF inhibitors also reduce phosphorylation of LDHA at the $Y^{10}$ residue. Phosphorylation of FRS2, an FGFR-interacting protein was included as a control. SDHA served as loading control. (F) Rate of lactate production from explants treated with DMSO (n = 5) or FGF inhibitors PD173074 (5 mM) (n = 6), PD173074 (20 mM) (n = 5), TKI258 (n = 6). (G) Steady-state ATP levels per retina in explants after culture with TKI258 or DMSO. The retinae were transferred to Krebs'-Ringer's with $NaN_3$ or NaCl (untreated group) for 30 min followed by harvest for ATP extraction. $n = 7$, DMSO+NaCl; $n = 9$, TKI258+NaCl; $n = 9$, DMSO+$NaN_3$; $n = 9$, TKI258+$NaN_3$. Data are Mean±SD. Statistics, Two-way ANOVA with Tukey's correction. (H) NADPH steady state levels in explants as a percentage of those measured in freshly isolated retina. Explants were treated with DMSO, oxamate, PD173074, TKI258 or left untreated in culture medium. n = 4 groups. Unpaired $t$-test with Kolmogorov-Smirnov correction for indicated pairs. (I) NADP steady-state levels in explants as a percentage of those measured in freshly isolated retina. Explants were treated with DMSO, oxamate, PD173074, TKI258 or left untreated in culture medium. Oxamate, n = 5; rest, n = 6 groups. Unpaired $t$-test with Kolmogorov-Smirnov correction for indicated pairs. (J) Blocking glycolysis or FGF signaling reduced EU incorporation in nascent RNA. Explants were treated with DMSO, oxamate, TKI258 or Actinomycin D (RNA Pol II inhibitor) followed by incubation with EU. (K) Quantitative PCR analysis of transcripts to ascertain relative expression of FGF or non-FGF targets (Arr3, Rs1) in explants cultured with or without RPE/Sclera complex (+RPE or –RPE respectively). (L) Ability to produce lactate from neural retina increased when cultured in the presence of RPE/Sclera complex (+RPE) (n = 11) as compared to those that were cultured without the complex (-RPE) (n = 9). Addition of FGF2 in –RPE cultures restored the ability (-RPE+FGF2) (n = 8). Retinal explants were cultured with RPE attached in the explant culture medium. Before transferring them to Krebs's-Ringer's for lactate estimation, the RPE/Sclera complex was removed and intact neural retina was used. For –RPE conditions, neural retina was cultured in explant medium followed by transfer to Krebs'-Ringer's. FGF2 was added to the explant culture medium but was absent in the Krebs'-Ringer's for -RPE+FGF2 condition. Data depict median in 1–99 percentile box and whiskers plot. Hinges extend between 25th to 75th percentiles. Statistics, Ordinary one-way ANOVA with Tukey's correction. ONL, outer nuclear layer. INL, inner nuclear layer. GCL, Ganglion cell layer.

The following source data and figure supplement are available for figure 4:

*Figure 4 continued*

**Source data 1.** Source data for *Figure 4F–I,K and L*.

**Figure supplement 1.** Model summarizing regulation of glycolysis and its contribution to photoreceptor physiology.

and phosphorylation at $Y^{105}$ was probed (*Figure 4C*). A light-dependent increase in phosphorylation at $Y^{105}$ of PKM2 was observed. Thus, this phosphorylation site might be one of the target sites for physiologically relevant signaling events regulating aerobic glycolysis. Light-dependent phosphorylation of this epitope in the retina has also been reported recently using IHC and immunoblot approaches (*Rajala et al., 2016*). The authors observed changes in signal intensity on IHC, that were dependent on light and activation status of the phototransduction pathway. Similarly, the $Y^{105}$ epitope showed light-dependent phosphorylation as assessed by immunoblot analysis of total retinal lysates. Our results on immunoprecipitated PKM2 confirm that this protein is among the targets of a light-dependent signaling pathway. Thus, phosphorylation of this site was then used as a proxy for the tyrosine kinase signaling pathways that could phosphorylate PKM2 in the retina. Freshly explanted retinae were cultured with antagonists targeting specific pathways: Afatinib (EGFR), Dasatinib (Src), BMS536924 (Insulin/IGF), PD173074 (FGFR1) and Dovitinib/TKI258 (FGFR1 and FGFR3). PKM2 was immunoprecipitated and its phosphorylation at $Y^{105}$ probed (*Figure 4D*). FGF inhibitors, PD173074 and TKI258, reduced PKM2 phosphorylation. Tyrosine kinase signaling can also target multiple nodes, including pyruvate dehydrogenase kinase and LDHA (*Fan et al., 2011*), and regulate aerobic glycolysis in cancer. We observed that treatment with either PD173074 or TK1258 also resulted in a dose-dependent decrease in LDHA phosphorylation at the $Y^{10}$ residue (*Figure 4E*). Thus, FGF signaling potentially targets multiple nodes in order to regulate aerobic glycolysis in the retina.

To determine if FGF signaling might regulate lactate production, freshly explanted retinae were cultured with TKI258 or PD173074, and lactate secretion was measured. Significantly reduced lactate secretion (*Figure 4F*) was seen to result from inclusion of either drug. In addition, inhibition of the FGF pathway resulted in increased mitochondrial dependence on ATP steady state maintenance (*Figure 4G*). Thus, one role for FGF signaling in the adult retina is to promote glycolytic reliance. FGF signaling is required for the maintenance of adult photoreceptors in mice and zebrafish (*Campochiaro et al., 1996*; *Hochmann et al., 2012*; *Qin et al., 2011*). Since it is possible that some of the effects are via regulation of aerobic glycolysis, we examined whether aerobic glycolysis promotes anabolism in the retina. Inhibition of aerobic glycolysis by oxamate treatment or by FGF inhibition resulted in significantly lower steady state NADPH levels- a key cofactor in biosynthetic pathways for lipids, antioxidant responses, and the visual cycle (*Figure 4H*). We also observed that interference with aerobic glycolysis did not result in an equivalent reduction in $NADP^+$ steady state levels (*Figure 4I*). The lowering of NADPH level could be attributable to attenuation of the PPP-shunt as a result of decreased glycolytic flux and/or increased usage of NADPH to quench the reactive oxygen species- an unavoidable consequence of increased mitochondrial dependence. We also assessed other effects on cellular anabolism. Nucleotide availability for nascent RNA synthesis was visualized using ethynyl uridine (EU) incorporation after treatment with oxamate and TKI258. Marked reduction in nascent RNA synthesis was evident following inhibition of LDH or FGF signaling (*Figure 4I*).

Among the large family of FGFs, basic FGF (bFGF/FGF2) has been the most studied in the adult retina. In adult mice and primates, FGF2 is localized to a matrix surrounding photoreceptors and/or is found on their OS (*Gao and Hollyfield, 1992*, *1995*; *Hageman et al., 1991*). The RPE might contribute to a high FGF2 concentration near photoreceptors via biosynthesis, and/or create a barrier to its diffusion from a retinal source. We first examined the role of the RPE in FGF-signaling. Adult retinal explants were cultured with the RPE/choroid/sclera complex, and expression of FGF target genes in the neural retina was compared with that of explants cultured without the attached complex. In the absence of this complex, the transcripts of known FGF signaling targets displayed reduced steady state levels (*Figure 4J*). To assess if the reduction in FGF targets was part of a general transcription downregulation or specifically due to dampened FGF signaling, we examined expression of retinoschisin (*RS1*) or cone arrestin (*Arr3*), genes expressed at moderate levels in the

retina (*Blackshaw et al., 2001*) (*Figure 4J*). These genes were not downregulated in the absence of the RPE complex.

The effect of the RPE complex on aerobic glycolysis was analyzed by quantifying lactate production (*Figure 4K*). Culturing retinae in the presence of the RPE complex resulted in a small, but significant, increase in the ability to produce lactate. Addition of FGF2 in the culture medium was sufficient to increase lactate production from explants cultured without the RPE. Together these data suggest that the RPE/choroid/sclera complex contributes to FGF signaling in the neural retina and that this signaling pathway plays a role in regulating the Warburg effect.

## Discussion

Several reports suggest that aerobic glycolysis is a feature of some normal proliferating somatic cells (*Agathocleous et al., 2012*; *Brand and Hermfisse, 1997*; *Wang et al., 2014b*; *Zheng et al., 2016*), and not just of cancer cells. Our work expands the cell types where aerobic glycolysis can occur to include a mature cell type, the differentiated photoreceptor cell. Like proliferating cells, rod photoreceptors utilize aerobic glycolysis to meet their anabolic needs. A critical aspect of aerobic glycolysis is its ability to be regulated. The data presented here suggest that allostery and FGF signaling are the regulatory mechanisms in the retina. We favor a model where aerobic glycolysis appears to be relevant to photoreceptors not only for organelle maintenance, but likely also helps photoreceptors meet their multiple metabolic demands (*Figure 4—figure supplement 1*).

In light of this model, it is important to assess the genetic tools that we employed to probe this pathway. Since we drove shRNA expression for *Ldha* and *Pkm2* knockdown from a constitutively active promoter (U6), we speculate that there could be an effect during retinogenesis. We reproducibly observed retinal thinning, indicated by a reduced number of nuclear rows (*Figure 1H*), especially in very well electroporated retinae. The thinning could be due to perturbation in the cell cycle of retinal progenitor cells, increased cell death, or both. As it is known that there is a role for LDHA and PKM2 in cell proliferation, it is quite likely that such an effect occurred here. The reduced retinal thickness was also apparent in some retinae from dark-reared animals that received the shRNA-encoding constructs against *Ldha* (*Figure 1J*). Many photoreceptors that received knockdown constructs against *Ldha* or *Pkm2* showed a significant increase in their OS length after dark rearing. This result argues for a physiological effect due to light exposure having an effect on the OS length, rather than a developmental defect. In addition, our experiments with *Tigar* gain-of-function, where expression is achieved in a spatiotemporal manner in order to have a minimal effect on retinal development, suggest that the effects of glycolytic perturbation on photoreceptor OSs can be parsed from the confounding effects on retinogenesis.

Aerobic glycolysis in the retina may have implications for blinding disorders. Studies on retinal degenerative disorders indicate that there are metabolic underpinnings to photoreceptor dysfunction, especially those centering around glucose uptake and metabolism (*Aït-Ali et al., 2015*; *Punzo et al., 2009*). Furthermore, reducing metabolic stress prolongs survival and improves the function of photoreceptors (*Venkatesh et al., 2015*; *Xiong et al., 2015*). In such treated retinae, there is a trend toward upregulation of glycolytic genes (*Venkatesh et al., 2015*) or metabolites (*Zhang et al., 2016*). However, a direct cause-and-effect relationship between cell survival and glycolysis has not been established. Our results highlight the metabolic strategies employed by healthy photoreceptors and provide a rational basis for the identification of candidate factors that would further clarify the role of glycolysis in retinal degeneration.

## Materials and methods

### Plasmids, viruses, in vivo electroporation and transfection

The synthetic promoter, CAG, consisting of cytomegalovirus (CMV) enhancer, chicken $\beta$-actin and rabbit $\beta$-globin gene splice acceptor was used for expression and genetic complementation. The expression pattern from this promoter when delivered by electroporation has been described previously (*Matsuda and Cepko, 2007*). Co-electroporation of a plasmid encoding myristoylated/membrane green fluorescent protein (mGFP) allowed visualization of cells that received the plasmid and marked the inner and outer segments. Co-electroporation rate of plasmids to the retina is close to

100% (*Matsuda and Cepko, 2007*). Full-length rat *Ldhb* (r*Ldhb*), human TIGAR, mouse PFKFB3 and human Pkm2 cDNA were obtained from Open Biosystems/GE Dharmacon. Subcloning, epitope tagging and site-directed mutagenesis were carried out by routine molecular biology procedures. For short hairpin (sh) design targeting PKM, and *Ldha* following resources/software were used: The RNAi consortium, CSHL RNAi central, iRNAi, Invitrogen Block-iT RNAi designer. Designed sh oligos were subcloned in pLKO.1 TRC backbone to be driven by the U6 promoter (Addgene, Cambridge, MA, #10878) and the sequences used in this manuscript are listed in *Supplementary file 2*. Four hairpin constructs were screened for *Ldha* and 72 were screened for PKM1/PKM2. Those hairpins that targeted specific mouse sequences but did not target human *Pkm2* were chosen. The murine FLAG-tagged PKM1 and PKM2 cDNAs were obtained from Addgene (#44240 and #42512) and subcloned in pCAG-EN. The pyruvate kinase activity from these ORFs has been already reported (*Anastasiou et al., 2011*).

The plasmids were mixed in equal molar ratios by accounting for their lengths and subjected to Phenol:Chloroform extraction followed by ethanol precipitation and resuspended to a final concentration of 1 mg/mL in Phosphate Buffered Saline. Subretinal in vivo injections and electroporation were carried out as described earlier (*Wang et al., 2014a*). When possible, the control and experimental constructs were injected in the pups of the same litter and the tail termini were snipped (or left uncut) to identify them later. For knockdown assays or testing expression from plasmids, transfection in HEK293T cells was carried out as using polyethylenimine (PEI). These cells were maintained as a lab stock and were subjected to periodic in-house testing for mycoplasma. Since, these cells were used for protein overexpression and knockdown analyses, concerns of misidentification do not apply to the current work and hence were not checked by third-party testing services.

For making the AAV-mGFP and AAV-TIGAR constructs, the CMV promoter in the empty AAV-MCS8 vector (Harvard Medical School DF/HCC DNA Resource Core) was replaced with the bovine rhodopsin promoter (*Matsuda and Cepko, 2007*). Woodchuck hepatitis virus posttranscriptional response element (WPRE) was added to enhance expression. Capsid type 8 AAVs were produced and titered as described previously (*Xiong et al., 2015*). For subretinal injections of AAV, ~$3.5 \times 10^6$–$5 \times 10^6$ particles (based on genome copies) per eye were used. P6 pups were injected in order to transduce cells after the proliferative phase of retinogenesis so as to minimize any detrimental effects on cell division and dilution of replication-incompetent viruses. The extent of infection was assessed with a Keeler indirect ophthalmoscope using the cobalt blue filter and Volk 78 diopter lens on non-anesthetized animals. Mice with edge-to-edge infection were tagged and used subsequently for lactate assays and immunoblotting.

## Mice and animal husbandry

Timed pregnant, wild-type CD1 female mice were obtained from Charles River Laboratories, Boston, MA, and P0-P1 pups thereof were used in electroporations. C57BL/6J and the two-color Cre reporter mouse *Gt(ROSA)26Sor^tm4(ActB-tdTomato,-EGFP)Luo*/J (referred to as mT/mG and described previously [*Muzumdar et al., 2007*]) were obtained from the Jackson Laboratories (JAX), Bar Harbor, ME. *Ldha^fl/fl* (*Wang et al., 2014b*), *Pkm2^fl/fl* (*Israelsen et al., 2013*), Rod-cre (*Le et al., 2006*) mice have been described before. Rod-Cre; *Ldha^fl/fl* and Rod-cre; *Pkm2^fl/fl* mouse lines were established. For experimentation, these mice were backcrossed with *Ldha^fl/fl* or *Pkm2^fl/fl* parents and Cre^+ and Cre^− F1 progeny were used to ensure equivalent allelic copies of the *Cre* transgene, minimum genetic difference and ease of age-matching by using the siblings. Animals were housed at room temperature with 12 hr light and 12 hr dark cycle. Light inside the cages in the room varied from 0 to 3 lx in the cage farthest from the light source to 300 lx in the cage closest to it. As a practice, the electroporated mice inhabited rack spaces where light intensity in the cages varied from ~175 to~235 lx. At weaning, the mice were segregated according to their sexes, thus a cage usually had the control and electroporated pups from the same litter. Tamoxifen injections were carried out as described previously (*Matsuda and Cepko, 2007*). For dark rearing, electroporated animals were raised with their mothers until P11, when the eyes started to open. Following this, they were transferred to animal housing maintained in darkness until weaning age, when they were weaned and group housed in dark until indicated times for harvest. In order to minimize effects due to circadian regulation of OS growth, the light and dark-reared animals from all the groups (control, LDHAsh and PKM2sh) were harvested on the same day and within 3 hr of each other. Water and chow were available *ad libitum*. Animal care was following institutional IACUC guidelines.

## Dissections and adult explant cultures

Wild-type, pigmented C57BL/6J mice (JAX) were used for explant cultures since presence or absence of RPE was easily discernable. For adult retinal cultures, P23-P28 animals were euthanized by CO2 asphyxiation and freshly enucleated eyes were dissected rapidly in Hanks buffered saline solution (HBSS) (Invitrogen, Carlsbad, CA). Extraocular tissue was trimmed off and the cornea and iris were carefully removed. Sclera along with the RPE was gently removed. This was done primarily for two reasons: (1) In our assays the presence of Sclera/RPE complex significantly reduced the efficacy of drug treatments and, (2) secreted lactate was not detected from freshly isolated eyecup with intact sclera. Lens was retained to keep the sphericity of the retina for uniform access to the medium. Explant medium consisted of Neurobasal-A, 0.2% B27 supplement, 0.1% N2 supplement, 0.1% Glutamax and penicillin/streptomycin (all Invitrogen). Retinae were incubated in freshly prepared explant medium constantly supplied with 95% $O_2$ + 5% $CO_2$ (Medical Technical Gases) at 37°C in a roller culture system (B.T.C Engineering, Cambridge, UK) for indicated times. At the end of incubation period, the lens was removed and the retinae were quickly rinsed with prewarmed Krebs' Ringers medium (98.5 mM NaCl, 4.9 mM KCl, 2.6 mM $CaCl_2$, 1.2 mM $MgSO_4$, 1.2 mM $KH_2PO_4$, 26 mM $NaHCO_3$, 20 mM HEPES, 5 mM Dextrose) saturated with 95% $O_2$. Retinae were again incubated in 0.5 mL Krebs' Ringers medium for 30 min in roller culture with 95% $O_2$ supplied. The supernatant and retinae were rapidly frozen separately at the end of the experiment. DMSO was used as vehicle control for water-insoluble solutes. Sodium oxamate or sodium azide was dissolved in the medium. Equimolar amount of sodium chloride was used as control for osmotic pressure, a colligative property. For +RPE experiments, the extraocular tissue was trimmed off, cornea and iris removed and the eyecups were incubated in the explant culture medium. At the end of the incubation, the RPE/sclera complex was removed along with the lens and the neural retina was incubated in the oxygenated Kebs's Ringers medium for 30 min as described earlier to assay secreted lactate. Thus, our experiments assess the effect of RPE/sclera complex on the ability to produce lactate by neural retina.

## Drugs

Sodium Azide (20 mM, Sigma-Aldrich), Sodium Oxamate (50 mM, Sigma-Aldrich), FX11 (10 µM, Calbiochem, San Diego, CA), BMS 536924 (5 µM, Tocris, Minneapolis, MN), Afatinib (5 µM, Selleckchem, Houston, TX), Dovitinib/TKI258 (5 µM, Selleckchem), Dasatinib (5 µM, Selleckchem), PD173074 (5 µM or 20 µM, Selleckchem), Actinomycin D (5 µM, Sigma-Aldrich, St. Louis, MO), FGF2 (2 µg/mL, Cell Signaling, Danvers, MA).

## Immunoprecipitation and immunoblotting

BL/6J retinae without RPE were homogenized in Lysis buffer (5 mM HEPES, 1 mM DTT, 1 mM ATP, 5 mM MgCl2, 1% glycerol, Complete Protease Inhibitor (Roche) and PhosStop phosphatase inhibitor (Roche, Basel, Switzerland). Immunoprecipitation was carried out using rabbit anti-PKM2 and rabbit IgG isotype control followed by sheep anti-rabbit-conjugated Dynabeads (Life Technologies). Immunoprecipitates were boiled and loaded on 10% SDS-PAGE gels followed by transfer on Hybond nitrocellulose membranes (GE Amersham, Amersham, UK). Membranes were blocked with 5% non-fat milk in 1X Tris Buffered Saline +0.1% Tween-20. A conformation-specific mouse-anti rabbit secondary and HRP-conjugated goat-anti-mouse (Jackson Immunoresrearch, 1:10,000) tertiary antibodies were used followed by Enhanced Chemiluminescent (ECL) detection using substrate from GE Amersham.

## Immunohistochemistry

Enucleated eyes were fixed overnight at 4°C in 4% formaldehyde. The eyes were passed through an increasing concentration of sucrose (5%, 15%, 30%) followed by equilibration in 1:1 30% sucrose: OCT (Sakura Finetek, Torrance, CA) and frozen on dry ice. Eighteen micron cryosections were cut using a Leica CM3050S cryostat. Antibodies used are listed in *Supplementary file 3*. Heat-mediated antigen retrieval at pH 8 was carried out. For HRP staining, Cell and Tissue staining kit (R and D systems) was used. Confocal images were acquired on Zeiss LSM710 or LSM780 inverted microscope. The intensity and pixel saturation were calibrated for inner and outer segment label (mGFP) so that details in these cellular features were retained. Thus, due to intense signal of mGFP in the outer

segments, the labeling in other cells of the inner retina seems variable and less bright despite electroporation known to target these cells. Images were processed on ImageJ. Maximum intensity projections are depicted. Colocalization was confirmed by individual merges of coplanar sections along the z-axis. For IS/OS length measurements, the orthogonal projections of sections were used. The projections spanning the entire IS/OS volume ensure changes due sectioning angle have a minimal effect. Multiple quantifications across the electroporated field were done for at least three retinae. Expression by IHC was confirmed in both CD1 (albino) and BL/6J (pigmented) mice. Sclera and RPE were preserved in electroporated eyes to ensure that outer segments were not ripped during the dissections. For all procedures involving antibodies, multiple antibodies were sourced and tested whenever possible (*Supplementary file 3*). Previously published antibodies were included and cross-verified with other commercially available antibodies. IHC data were always cross-verified with RNA ISH.

## In situ hybridization

In situ hybridization was carried out as described earlier (*Blackshaw et al., 2001*). Probe sequences are presented in *Supplementary file 4*. For *Pfkfb1, Pfkfb2, Pfkfb4, Srsf3* and *Ptbp1*, tyramide amplification (Perkin, Waltham, MA) was used. Bright-field images were acquired on Nikon Eclipse E1000 microscope.

## ATP, Lactate and NADPH assay

For ATP estimation, individual retinae were rapidly frozen in liquid nitrogen at the end of the assay. ATP was measured using ATP bioluminescence kit CLS II (Roche/Sigma-Aldrich). For secreted lactate estimation the retinae were incubated in Krebs' Ringers medium after indicated treatments. The supernatant from above was used with Lactate assay kit (Eton Bioscience, San Diego, CA). Amount of lactate produced in 30 min was assayed. Intracellular lactate was estimated for AAV-transduced retinae because a large number of mice had to be injected and screened for complete, edge-to-edge infection. Thus, infected retinae at specific age were harvested and frozen as they became available. All these retinae were harvested 4–5 hr after lights were turned on in the facility to minimize variability due to possible cyclical diurnal changes in metabolism. Two to three retinae were pooled into a group and frozen together. Five such groups (*n* = 5) were used for assaying lactate after AAV-mGFP and AAV-TIGAR infection. The retinae were homogenized with the Lactate Assay buffer (Fluorometric Lactate Assay kit, abcam, Cambridge, MA). A small aliquot was removed for protein estimation and subsequent immunoblotting and the remainder was passed through 10 kDa protein filtration column (abcam) to remove proteins and thus minimize interference due to endogenous lactate dehydrogenase in the lactate assay. For protein estimation, Qubit protein assay (Invitrogen) was used since it is not affected by the presence of detergents in the Lactate Assay buffer. NADP and NADPH was assayed using Fluoro NADP/NADPH kit (Cell Technology, Fremont, CA) following manufacturer's instructions. The quantifications for NADP and NADPH were made separately and thus represent different retinae and treatments.

## COX and SDH histochemistry

Histochemistry on fresh and unfixed retinal tissue was carried as described earlier for brain tissue (*Ross et al., 2010*). The assay relies on the ability of functional cytochrome oxidase to catalyze oxidative polymerization of 3,3'-diaminobenzidine (DAB) (an electron donor) to brown indamine product. Succinate dehydrogenase assay is based on the ability of this enzyme to oxidize supplied succinate and in turn reduce a ditetrazole (NBT) to dark blue diformazan using phenazine methosulfate (PMS), an intermediate electron carrier.

## 5-Ethynyl uridine (EU) labeling

Explants were cultured with indicated drug or DMSO for 5 hr followed by 1 mM EU (Life Technologies) with the drug for additional 2.5 hr. The retinae were fixed, cryosectioned and processed for label detection using Click chemistry reagents (Life Technologies).

## Quantitative RTPCR

RNA was isolated using TRIzol reagent (Life Technologies, Carlsbad, CA) from 3 to 4 retinae. Two µg RNA was subjected to cDNA synthesis using SuperScript III reverse transcriptase and random hexamers. QPCR was performed using power SYBR Green PCR Master mix (Applied Biosystems, Foster City, CA) on a 7500 Fast Real-Time PCR System (Applied Biosystems). Primer sequences are provided in *Supplementary file 5*. *Rpl13a* was used as internal reference and freshly isolated retinal tissue was used as calibrator sample. Expression ratio was calculated using $2^{-\Delta\Delta Ct}$ method. For each target gene, three technical replicates were simultaneously assayed to arrive at the average value for a biological replicate. Mean of three biological replicates was used to derive the $C_t$ value of each target.

## Rod isolation and cDNA synthesis

P0 CD1 mice were electroporated with Rho-dsRed plasmid which encodes for dsRed, driven by bovine rhodopsin promoter, which results in retinas with patches of dsRed expression only in rod photoreceptors (*Matsuda and Cepko, 2004*). Once they reached adulthood, mice were then euthanasized via $CO_2$ asphyxiation and the retinas were rapidly removed. The retinas were incubated for 5 min at 37°C in Hank's Balanced Salt Solution (HBSS) supplemented with 10 mM HEPES and 5 mM EDTA and then gently triturated with a P1000. The dissociated retina was allowed to settle on sylgard-coated petri dishes. Rods expressing the dsRed reporter were identified by their red fluorescence using an inverted microscope and hand-pipeted directly into lysis buffer, and their cDNA amplified using the previously described protocol that utilizes oligo dT priming (*Goetz and Trimarchi, 2012*).

## Data collection and statistics

Data collection was from non-randomized experiments. The primary experimenters were not blinded to treatments. No statistical methods to predetermine sample size were employed. No assumptions for potential outliers were made and hence all data points were included in analyses and depicted. Normality of data distribution was tested using D'Agostino-Pearson omnibus test. Non-parametric statistics were used when Gaussian distribution of data points could not be obtained. *p*-value denoted as: Not significant (NS), p>0.05; *p≤0.05; **p≤0.01; ***p≤0.001; ****p≤0.0001.

## Acknowledgements

We thank Ryan Chrenek, Lucy Evans, Parimal Rana, Lillian Horin and Alexandra McColl-Garfinkel for technical help. We are grateful to Will Israelsen and Matthew Vander Heiden for help with *Pkm2fl/fl* mice, Ying-Hua Wang and David Scadden for *Ldhafl/fl* mice and Yun Z Le for Rod-cre mice. We are indebted to Barry A Winkler for generous input on retinal metabolism, especially at the earlier stages of this work. We thank Ben Stranges (George Church lab) and Quentin Gilly (Norbert Perrimon lab) for access to the microplate readers. This work has been supported by the National Institutes of Health grant R01 EY023291 and the Howard Hughes Medical Institute.

## Additional information

### Funding

| Funder | Grant reference number | Author |
| --- | --- | --- |
| National Institutes of Health | R01 EY023291 | Yashodhan Chinchore<br>Tedi Begaj<br>David Wu<br>Eugene Drokhlyansky<br>Constance L Cepko |
| Howard Hughes Medical Institute | | Constance L Cepko |

The funders had no role in study design, data collection and interpretation, or the decision to submit the work for publication.

## Author contributions
YC, Conceptualization, Formal analysis, Investigation, Visualization, Methodology, Writing—original draft, Writing—review and editing; TB, DW, Methodology; ED, Validation, Methodology; CLC, Conceptualization, Resources, Formal analysis, Supervision, Funding acquisition, Writing—review and editing

## Author ORCIDs
Yashodhan Chinchore, http://orcid.org/0000-0003-3521-9291
Constance L Cepko, http://orcid.org/0000-0002-9945-6387

## Ethics
Animal experimentation: Animal care and use adhered to the Harvard Medical School's IACUC guidelines. Animals were handled in accordance with the protocol# 0428 and 04537.

# Additional files

## Supplementary files
• Supplementary file 1. qPCR analysis of target genes in isolated rod samples.

• Supplementary file 2. shRNA-encoding constructs used and targeted regions in the cDNA.

• Supplementary file 3. List of antibodies.

• Supplementary file 4. Probe sequences for in situ hybridization.

• Supplementary file 5. Primers for qPCR analysis.

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
