## [Decision Letter]

Thank you for submitting your article "Glycolytic reliance promotes anabolism in photoreceptors" for consideration by *eLife*. Your article has been favorably evaluated by Richard Aldrich (Senior Editor) and three reviewers, one of whom, Jeremy Nathans, is a member of our Board of Reviewing Editors. One of the reviewers (Reviewer 1) has agreed to reveal his identity: James B. Hurley.

As you will see, all of the reviewers were impressed with the importance and novelty of your work. I am including the three reviews (lightly edited) at the end of this letter, as there are a variety of specific and useful suggestions in them. Some of the comments can be handled with changes in the text or responses to the reviewers but others require additional experimental work. It may help us to assess your prospects for a timely revision if you write back with responses and a plan to address the concerns of the reviewers.

*Reviewer #1:*

This outstanding, highly significant, timely and well written report aims to connect glycolytic metabolism in retinal photoreceptors with their size. The investigators link recent findings in the field of cancer metabolism to the influence of metabolism on growth and survival of primary sensory neurons. They use sophisticated and appropriate genetic approaches to perturb the normal metabolic functions of photoreceptors in vivo. They show that deficiencies in aerobic glycolysis alter photoreceptor growth. This is a major advance for understanding what is needed for homeostasis in primary sensory neurons. The authors should address the following specific comments.

1) Introduction, second paragraph: Please note that Wang et al., (1997) PMID: 9179314 showed that aerobic glycolysis in the retina originates in the outer retina.

2) For the experiments in Figure 1 please specify concentrations of oxamate and azide.

3) This statement is confusing: "A tetramer of all ldha subunits has high affinity for pyruvate and a higher Vmax for pyruvate reduction to lactate." Is this referring to a higher Vmax than ldhb or a higher Vmax for lactate to pyruvate? Also, it isn't clear from this which one has a higher affinity for pyruvate, Ldha or Ldhb tetramers? Please spell out the kinetic differences between the LDH isoforms in more detail.

4) The legend for supplement Figure 1—figure supplement 2 should state the conclusion of that experiment – that the Cre is not expressed in cones.

5) I'd think it would be helpful for the authors to compare more comprehensively the top two rows of panel H of Figure 1. It looks like the overall thickness of the retina is smaller in the second row. Is that real and reproducible?

6) How do the authors correct for variations in the angle of sectioning when measuring the lengths of the OS? It would be useful for at least a few of the experiments to present the OS and IS length as a% of the total thickness of the retina. Also, are the morphologies of the Muller cell or RPE affected?

7) For the electroporation of shRNA experiments for LDHA the authors highlight how electroporation occurs in only 15-30% of the retina. That means there must be some regions where there is a transition from high to low LDHA expression. Can the authors tell whether or not the loss of LDHA is cell autonomous or is the length of the outer segment influenced by the amount of lactate produced by its neighboring rods? Using anti-rhodopsin antibodies (see next comment) to label the OS in IHC may help reveal the phenotype in these regions.

8) The authors use mGFP expression to measure the OS and IS length. Would it be possible/worthwhile to confirm a couple of the findings by simple staining some of the sections with a rhodopsin antibody that would very clearly label the OS and then quantifying the lengths of the OS to make sure that type of measurement is consistent with the measurements made from the expressed mGFP? Please either address this experimentally or explain why it would not be practical or necessary.

9) For the experiments in Figure 1 and also other experiments where outer segment and inner segment length were measured – were all the retinas collected at the same time of day?

10) Maybe I overlooked it, but I could not find the part of the text that refers to Figure 2—figure supplement 1 panels C-F.

11) Subsection “Nonequivalent roles of pyruvate kinase isoforms”, first paragraph: Please also cite recent paper from the Rajala lab (PMC5121888) that show expression patterns of PKM1 and PKM2 and tyrosine phosphorylation of PKM2. Also please cite the relevant IOVS paper by Casson et al. PMID: 26780311. Another paper that should be cited regarding PKM2 in the retina is Molecular Vision paper by Rueda et al. (PMID: 27499608)

12) The difference in the overall thickness of the retina is particularly obvious in Figure 3—figure supplement 1 (the rescue with huPKM2). It seems like the whole retina is affected by the PKM2 expression. Is that correct? The authors should address the possibility that the whole retina is affected in a non-cell autonomous way. (Minor note: the legend refers to the boom panel on the right but it is on the left).

13) Subsection “Nonequivalent roles of pyruvate kinase isoforms”, fourth paragraph: PKM2^fl/fl^ and Rod-cre mice were used to knockout PKM2 in rods for the lactate measurements. Please note whether the rods were shorter in these experiments. Also, if I'm right that the overall thickness of the retina changes do that also occur in the PKM2^fl/fl^/rod-cre retinas?

14) In the subsection “Fibroblast growth factor signaling regulates anabolism”, the authors should cite a recent paper, PMC5121888, that also confirms that Y105 of PKM2 is phosphorylated in mouse retinas.

15) Please include MW marker positions on Figure 4. Also, please explain what the "isotype-matched" means.

16) The authors should be careful to not over-interpret data in some of their descriptions. For example, the authors state "Impact of glycolytic perturbation on nucleotide availability was directly visualized[…]". Since FGF signaling can influence other processes besides glycolysis I think this is over-interpreting the data. It would be better to just say that FGF receptor signaling influences nucleotide availability and this could be linked to limitation of glycolysis. Alternatively the connection could be strengthened by evaluating the effects of PKM2 or LDHA inhibition on EU incorporation.

*Reviewer #2:*

Vertebrate photoreceptors are among the most metabolically active cells, exhibiting a high rate of ATP consumption. This is coupled with a high anabolic demand, necessitated by the diurnal turnover of a specialized membrane-rich organelle, the outer segment, which is the primary site of phototransduction. It is not clear to date that how photoreceptors balance their catabolic and anabolic demands. The current study has attempted to address this fundamental issue in photoreceptor biology. The authors have used several biochemical, immunological, genetic and viral transduction methods to address the importance of glycolysis on the outer segment biogenesis. The study is interesting but failed to support the authors claim that glycolysis regulates the outer segment biogenesis.

1) In these studies, authors have examined the isoform expression of LDH, PKM2, then phosphorylation state of PKM2 under dark- and light-adapted conditions, identified FGF signaling promotes PKM2 phosphorylation and splicing regulation of PKM1 and PKM2. Some of these studies have been done before by other labs (see below), which the authors did not acknowledge or reference in their manuscript. These include:

• Identification of LDH isoforms (Casson RJ et al. 2016 IOVS).

• PKM1 and PKM2 isoform characterization, light-dependent tyrosine phosphorylation of PKM2 (Rajala et al. 2016 Sci Rep).

• The authors claimed in this study that fibroblast growth factor (FGF) signaling was found to regulate glycolysis through phosphorylation of PKM2. This finding is not novel as it has been elegantly shown in tumor cells that FGFR regulates PKM2 phosphorylation (Hitosugi et al. 2009 Sci Signal).

• Regulation of PKM1 and PKM2 splicing (Su et al. (2017 Mol Cell Biol)

The authors are urged to cite these earlier references and give proper credit for these studies. They could discuss how the published results are similar or differ from their observations in this manuscript.

2) LDH isoforms identification has previously been reported (Casson RJ et al. 2016 IOVS). The authors must cite this manuscript.

3) The authors stated in the manuscript (subsection “Lactate producing isoform of Ldh in photoreceptors”, last paragraph) the recombination efficiency with rod-cre varied between 50-90%, but that is not correct. The rod-cre used in these studies will not recombine more than 50%. The authors have shown only protein expression by Western blots. They need to show the deletion by immunohistochemistry.

4) The authors have shown that lactate production was significantly reduced in conditional LDH-A mice. Why did the authors not study OS biogenesis in LDH-A deleted mice? The shRNA strategy is not well justified. Have the authors examined the OS in LDH-A KO mice? Generally, the shRNA approach may not knock-down completely the gene of interest but conditional deletion will? The authors observed the shortening of OS. Could this be an off-target effect? LDH-A is also expressed in other layers of the retina (INL and IPL).

Immunohistochemistry is not the ideal way to demonstrate OS length; hence the authors measured IS+OS. The authors should use ultrastructural studies, such as EM or high resolution LM to demonstrate the OS length phenotype. Some of the micrographs show thinning of the outer nuclear layer thickness (e.g., DAPI stained sections in Figure 1, Figure 2, and 3C, suggesting retinal degeneration in these genetically modified retinas, which could argue against shortening of OS length. Did the authors do TUNEL or any other test for dying photoreceptor cells?

5) Could shRNA knock down in other retinal layers (may be Muller cells) may indirectly affect the structure of OS?

6) Figure 1 – There is no evidence of disc shedding in this experiment. Could there be less opsin trafficking to the OS? Such a possibility cannot be ruled out. This experiment is overstated.

7) – The statement "lactate production by the photoreceptors cannot be attributed to lack of mitochondrial activity." The authors have done experiments on (Figure 1) wild- type retinas. Have they carried out these experiments in LDH-A-knockdown or KO mice?

8) Figure 2 – what is the rationale to regulate TIGAR expression spatially and temporarily? In Figure 2 the authors used AAV-mediated expression of TIGAR. Lactate levels were done in AAV-TIGER (Figure 2) but not for Figure 2? It is very confusing, and there was no rationale provided for these experiments. It seems that authors may have difficulty in measuring lactate levels for the inducible expression system?

9) Figure 3 – PKM1 and PKM2 expression has recently been reported (Rajala 2016 Sci Rep). The authors have not cited this reference.

10) The authors have shown the developmental expression of PKM2 and PKM1 on western blots (Figure 3), which is not the ideal way to show the developmental expression. If the authors wanted to show this, they should provide immunohistochemistry or ISH.

11) The authors used rod-cre to delete PKM2 and measured LDH activity (Figure 3). For structural studies, they used shRNA and examined OS length (Figure 3). These studies are not convincing. Why did they not observe similar shRNA effects with conditional PKM2 KO mice?

12) There is no indication of how much PKM2 is deleted or knocked down. In the absence of these experiments, it is very difficult to interpret the data. Moreover, the authors did not carry out any functional studies, such as ERG to examine the role of PKM2 in photoreceptor functions?

13) Figure 3—figure supplement 2 does not add any new information. The authors' data show opposite expression of these splicing factors. There was a study recently published showing that RBM4 Regulates Neuronal Differentiation of Mesenchymal Stem Cells by Modulating Alternative Splicing of Pyruvate Kinase M (Mol Cell Biol 2017).

14) The authors stated that PKM2 deletion upregulates PKM1, but has no effect on photoreceptor structure (Figure 3—figure supplement 1). On the other hand, forceful expression of PKM1 had a reduction in the length of OS? How do authors explain this discrepancy?

15) Figure 4 – FGF signaling – Authors have identified that FGF signaling promotes the phosphorylation of PKM2. It is not a novel finding. It has been shown in tumor cells (Hitosugi et al., 2009). The authors have not acknowledged this information in the current manuscript.

16) Figure 4 – PKM2 undergoes a light-dependent tyrosine phosphorylation on Tyr105. These studies have recently been reported by Rajala et al. 2016 (Sci Rep). The authors have not acknowledged this study.

*Reviewer #3:*

This manuscript provides novel and interesting data on the reliance of aerobic glycolysis for photoreceptor outer segment renewal. Overall, the paper is very good and a significant contribution. However, there are some significant problems that need addressing before the results and conclusions that are presented can be accepted. In addition, there are several additional items that are off putting and overstepping the presentation and results.

First, the authors are only addressing aerobic glycolysis in the rod inner segments and outer segments. There does not appear to be any data for cones or rod and cone photoreceptor synapses. So, the title should reflect this and state something akin to "Aerobic glycolytic reliance promotes anabolism in rod photoreceptor outer and inner segments."

Second, the authors should provide more detailed and important information in the Methods. This information is critical for proper interpretation of the results. For example, what the light level in the animal room and cages, what time of day were the mice sacrificed, what area and quadrant of the retina was quantified for OS-IS measurements, how was this location established, what are the measurements for OS and IS alone as they could both change, how many non-adjacent sections per retina were quantified per mouse, what was the volume of incubation for the lactate assay, what type of confocal was used and how many Z-stacks were there per image, what was the exact age of mice used for the histology as in the subsection “Dissections and adult explant cultures”, the authors stated that they used P23-P28 mice. According to LaVail 1973 J Cell Biol, the outer segments in the C57BL/6J mice reach their rate of synthesis and disposal at around P21-25.

Third, the authors conclude from looking at their retinas 42-45 days following electroporation that the shortened rods were due to decreased synthesis due to decreased aerobic glycolysis. This cannot be accepted at face value without conducting two additional experiments: show rod disc synthesis results [e.g., per Young and Bok studies] and rod phagocytosis results [e.g., per LaVail studies].

Fourth, please explain why eye opening occurred on day 11 in these studies. Most published mouse paper find that this event occurs at postnatal day 14 +/- 1 day.

Fifth, the choice of words in several places throughout the manuscript are overstated or not properly used. This often results from lack of citation of previously conducted work. For example, they state that "The cell types that carry out aerobic glycolysis in the normal adult retina have not been determined." This is just false. The work of Rueda et al. published in Mol Vis in 2016 that shows cell-type glycolysis, aerobic glycolysis, high energy transferring kinases and oxidative phosphorylation in over 20 different compartments and cells in the retina. A summary figure clearly shows all of this data. However, the authors have not cited this manuscript anywhere in their manuscript.

For example, they state that "surprisingly the steady state levels of ATP…did not differ from controls." Winkler in J Gen Physiol 1981 showed that retinal ATP levels are steady under most conditions: not surprisingly since Lowry and co-workers demonstrated in the 1960s that this energy measure is the last to change even after anoxia.

For example, in the subsection “Aerobic glycolysis in the retina”, the authors incorrectly stated that adenylate kinase (AK) synthesizes ATP. The do not acknowledge that mouse retina also expresses creatine kinase (CK). The authors need to state that CK and AK serve to regenerate the ATP by reversible reactions that respond to the law of mass action. The implication of these enzyme reactions need to be explained as these enzymes may respond under conditions of high ATP hydrolysis or raising [ADP] (E.g., see the work of Wallimann, Linton et al. 2010; Rueda et al., 2016).

For example, the authors state that "presumably as a result of less shedding". This results from the process of "photostasis' as published in Exp Eye Res 1n 1986 by Penn and Williams.

For example, the authors state that "Our work expands the cell types where aerobic glycolysis can occur to include a mature cell type, the differentiated photoreceptor cell". As noted, the authors did not cite the prior work of Rueda et al. published in Mol Vis in 2016. This work also pyruvate competition in LDH histochemistry assays and confocal to demonstrate that Ldh-A was preferentially located in photoreceptors…not cited in the first paragraph of the subsection “Lactate producing isoform of Ldh in photoreceptors”, as confirmatory. So the current manuscript "confirms and definitely expands" would be the correct terminology.

For example, the Abstract says that the process of photoreceptor catabolism and anabolism is "poorly understood". It should be more precise as there are many studies on these process, but not in the context of rod outer segment biosynthesis.

Sixth, no retinal references are provided for COX and SDH histochemistry. Several previous papers conducted these experiments in rodents (SDH in rat: Hansson, Exp Eye Res 1970: COX in rat: Chen et al. Acta Ophthalmol 1989; COX in mouse: Rueda et al., Mol Vis 2016). The latter paper in adult mouse was not cited in the subsection “Functional mitochondria in photoreceptors”, although quantitative layer-by-layer COX activity was measured and presented. The authors stated that SDH activity was not different between inner and outer retina, but this does not agree with prior published results.

Seventh, the authors state that "PKLR transcripts were not detected in retina (data not shown)". However, they were found in the retina by other investigators (see Figure 1 of Rueda et al. Mol Vis 2016). No comment is made to acknowledge the difference.

Eighth, the authors should have used a higher resolution microscopy to show specifically the changes observed in the outer segments and possible exclude or determine the possible changes in inner segments and inner segment-mitochondria due to loss of aerobic glycolysis.

Ninth, please clarify in the first paragraph of the subsection “Allosteric regulation of glycolysis in photoreceptors”, the three listed criteria points.

Tenth, Ross et al. 2010a should be 2010 as there is no "b".

---

## [Author Response]

*Reviewer #1:*

*This outstanding, highly significant, timely and well written report aims to connect glycolytic metabolism in retinal photoreceptors with their size. The investigators link recent findings in the field of cancer metabolism to the influence of metabolism on growth and survival of primary sensory neurons. They use sophisticated and appropriate genetic approaches to perturb the normal metabolic functions of photoreceptors in vivo. They show that deficiencies in aerobic glycolysis alter photoreceptor growth. This is a major advance for understanding what is needed for homeostasis in primary sensory neurons. The authors should address the following specific comments.*

*1) Introduction, second paragraph: Please note that Wang et al., (1997) PMID: 9179314 showed that aerobic glycolysis in the retina originates in the outer retina.*

The work by Anders Bill and colleagues in the aforementioned publication (Wang et al., 1997, Acta Physiol. Scand., 160, 75-81) explored the production of lactate and consumption of oxygen and glucose in dark and light in the pig retina. They used direct measurement of metabolites (lactate and glucose) and oxygen from the vorticose vein. These vessels also drain from the uvea including anterior uvea. It is known that the lens cells are glycolytic after they expunge much of their organelles. Though light should not affect the glycolytic state of these cells, it is still an assumption until proven otherwise. The other effect that should be considered is that of the effect of constriction of sphincter pupillae muscles following light stimulation. How much lactic acid is produced from these muscles in their contracted state and how much do they contribute to the free lactate in the serum? Even if the contribution towards the lactate pool from the outer retina proves significant over that of the anterior uvea, can one be certain that it is attributable to photoreceptors? One can also interpret these results in the light of the lactate shuttle hypothesis i.e. Mueller glia secrete lactate (the directionality of the flow can be explained by consumption of lactate by the photoreceptors). Given that photoreceptors have an increased energy demand in the darkness, they consume more of the lactate, resulting in less being secreted into the blood stream. This would align with the observation of Wang et al. as they note ~60% decrease in lactate levels in darkness. Lastly the contribution of RPE to the steady state free lactate levels in the blood is not known, but should be considered when retinal physiology models are constructed. We do not contest the validity of their observations. But from our point-of-view, these data do not help to discriminate between two opposing models of photoreceptor physiology based on glycolysis and lactate production or consumption. Since that does not change our conclusion laid out in the second paragraph of the Introduction, that photoreceptors are assumed to rely on aerobic glycolysis and a systematic analysis of relative propensities of photoreceptors and other cell types, especially Mueller glia, for aerobic glycolysis has not been carried out, we had refrained from citing this work in the interest of space and number of articles that should ideally be referenced. Inclusion of the caveats listed above would have taken up quite a bit of space and we did not want to be overly critical of the previous work in this area by pointing these out. Incidentally, these caveats also apply to many such studies wherein the entire eye, or the entire retina, or the retina and associated tissues (e.g. RPE) were assayed.

*2) For the experiments in Figure 1 please specify concentrations of oxamate and azide.*

The concentrations of these agents and others are reported in Materials and methods (subsection “Drugs”).

*3) This statement is confusing: "A tetramer of all ldha subunits has high affinity for pyruvate and a higher Vmax for pyruvate reduction to lactate." Is this referring to a higher Vmax than ldhb or a higher Vmax for lactate to pyruvate? Also, it isn't clear from this which one has a higher affinity for pyruvate, Ldha or Ldhb tetramers? Please spell out the kinetic differences between the LDH isoforms in more detail.*

Given that our approach relied on exploiting relative differences between Ldha and Ldhb isoforms, we agree that a clearer explanation is warranted. We have made the suggested changes (subsection “Lactate producing isoform of LDH in photoreceptors”, first paragraph).

*4) The legend for supplement Figure 1—figure supplement 2 should state the conclusion of that experiment – that the Cre is not expressed in cones.*

We had refrained from stating conclusions in the figure legends, as we prefer that data in a figure show the conclusion. To be consistent, we are avoiding it here as well. We have stated the conclusions in the text body (subsection “Lactate producing isoform of LDH in photoreceptors”, last paragraph).

*5) I'd think it would be helpful for the authors to compare more comprehensively the top two rows of panel H of Figure 1. It looks like the overall thickness of the retina is smaller in the second row. Is that real and reproducible?*

We agree with the reviewer’s observation that the thickness of the retina is reduced. We have now addressed this issue in the Discussion (second paragraph).

*6) How do the authors correct for variations in the angle of sectioning when measuring the lengths of the OS? It would be useful for at least a few of the experiments to present the OS and IS length as a% of the total thickness of the retina. Also, are the morphologies of the Muller cell or RPE affected?*

We agree with the Reviewer’s concern that quantification involving OS length should factor in all the sources of variation. Changes in sectioning angle can introduce significant variability especially in the photoreceptors lying at the cut surface (closest to the imaging objective). We have addressed this issue in Materials and methods subsection “Immunohistochemistry”. Confocal imaging of the electroporated photoreceptors and orthogonal projections of photoreceptor volume along the Z-axis, enabled us to sample greater depth and image more photoreceptors in their entirety in the retinal tissue, rather than only those on the cut surface. We sampled multiple photoreceptors in any given field, multiple fields in an electroporated patch, and multiple retinae that were independently sectioned. In addition, we included all data points without setting statistical cutoffs for outliers (Materials and methods subsection “Data collection and statistics”). We also depict all the data points to give readers the idea of the data spread. In summary, we have made efforts to minimize variability in our data collection and include all data points for statistical analysis.

*7) For the electroporation of shRNA experiments for LDHA the authors highlight how electroporation occurs in only 15-30% of the retina. That means there must be some regions where there is a transition from high to low LDHA expression. Can the authors tell whether or not the loss of LDHA is cell autonomous or is the length of the outer segment influenced by the amount of lactate produced by its neighboring rods? Using anti-rhodopsin antibodies (see next comment) to label the OS in IHC may help reveal the phenotype in these regions.*

This comment and the following one addressed together as 1 response.

*8) The authors use mGFP expression to measure the OS and IS length. Would it be possible/worthwhile to confirm a couple of the findings by simple staining some of the sections with a rhodopsin antibody that would very clearly label the OS and then quantifying the lengths of the OS to make sure that type of measurement is consistent with the measurements made from the expressed mGFP? Please either address this experimentally or explain why it would not be practical or necessary.*

We have included rhodopsin staining for the *LDHA* knockdown retina (Figure 1—figure supplement 3). Due to the predominance of rods and abundance of rhodopsin, staining of outer segments using such methods usually results in an overwhelming signal that confounds resolution of individual cells and their features of interest (here outer segments). Since we had to label the electroporated cells anyway, we directly quantified the IS+OS lengths of mGFP^+^ photoreceptors. However, the reviewer’s suggestion is useful in highlighting the cell-autonomous nature of our perturbations. We do not observe a decrease in the length of rhodopsin-positive outer segments of photoreceptors that did not receive the plasmid (mGFP^-^ photoreceptors in the electroporated patch) when compared to outer segments outside the electroporated patch. We thus concluded that reduction in OS length of mGFP^+^ photoreceptors was due to a cell autonomous effect. One would expect that an effect on Mueller glia or the RPE would have affected the non-electroporated rods in the patch as well. In addition, this also addresses the reviewer’s concern that a cell receiving the knockdown construct does not result in an adverse phenotype on its nonelectroporated neighbor. Besides this suggested experiment, we are also presenting phenotypes observed using sparse electroporation of the retina, which is achieved by diluting the plasmid. Fewer cells are electroporated in this case, and again we observe a reduction in outer segments of electroporated photoreceptors.

Changes in the manuscript: Added Figure 1—figure supplement 3. The text is amended to describe this figure (subsection “Active LDHA supports outer segment biogenesis”, first paragraph). Added Figure 3—figure supplement 3 with amends to the text body (subsection “Nonequivalent roles of pyruvate kinase isoforms”, second paragraph) for sparse electroporation. Description of rhodopsin antibody is included in [Supplementary-material SD7-data].

*9) For the experiments in Figure 1 and also other experiments where outer segment and inner segment length were measured – were all the retinas collected at the same time of day?*

We appreciate pointing out this aspect to us. We did consider a possibility of a circadian/diurnal effect on the OS length. The retinae from six sets of animals: control (light and dark), LDHAsh (light and dark), PKM2sh (light and dark) were harvested in the first half of the day and within 3 hours of each other. Similarly, efforts were made to collect the AAV-TIGAR and AAV-mGFP infected retinae for lactate analysis at the same time of the day, as they became available. The timing was inadvertently not incorporated in the earlier draft. We have now updated the Materials and methods sections (” Mice and animal husbandry” and “ATP, Lactate and NADPH assay”).

*10) Maybe I overlooked it, but I could not find the part of the text that refers to Figure 2—figure supplement 1 panels C-F.*

The text referring these specific figure panels is in the second, third and fourth paragraphs of the subsection “Allosteric regulation of glycolysis in photoreceptors”.

*11) Subsection “Nonequivalent roles of pyruvate kinase isoforms”, first paragraph: Please also cite recent paper from the Rajala lab (PMC5121888) that show expression patterns of PKM1 and PKM2 and tyrosine phosphorylation of PKM2. Also please cite the relevant IOVS paper by Casson et al. PMID: 26780311. Another paper that should be cited regarding PKM2 in the retina is Molecular Vision paper by Rueda et al. (PMID: 27499608)*

These references are now cited.

12) The difference in the overall thickness of the retina is particularly obvious in Figure 3—figure supplement 1 (the rescue with huPKM2). It seems like the whole retina is affected by the PKM2 expression. Is that correct? The authors should address the possibility that the whole retina is affected in a non-cell autonomous way. (Minor note: the legend refers to the boom panel on the right but it is on the left).

The reviewer is correct in pointing out that all the retinal layers are affected, including the INL. Though that panel was more towards the periphery (which accentuated the retinal thinning), we have replaced it with a panel that lies in the central retina, closer to that of the rescue panels. We have observed this phenotype (thinning of ONL and INL) reproducibly in many different retinae that get well-electroporated. PKM1+2sh knocks down both PKM1 and PKM2 and is more effective in knocking down PKM2 than the PKM2sh (see the western blots in Figure 3—figure supplement 1, Panels D and E in the revised draft). Thus it is likely that this construct could perturb retinogenesis to a greater extent and/or can also affect the cells in the INL (that express PKM1). The genetic sufficiency of PKM2 in restoring the OS length is key and forms an important aspect in concluding the nonequivalent roles of PKM1 and PKM2 in the retina. We appreciate the reviewer’s comments on the labeling of panels. We have now corrected this in the figure legend.

*13) Subsection “Nonequivalent roles of pyruvate kinase isoforms”, fourth paragraph: PKM2^fl/fl^ and Rod-cre mice were used to knockout PKM2 in rods for the lactate measurements. Please note whether the rods were shorter in these experiments. Also, if I'm right that the overall thickness of the retina changes do that also occur in the PKM2^fl/fl^/rod-cre retinas?*

We are including the panel for rhodopsin staining to label the rod outer segments of Rod-cre> PKM2^fl/fl^ in Figure 3—figure supplement 2 (Figure 3—figure supplement 1 in earlier version). Fortunately, the mosaic nature of recombination due to the Rod-cre line enables us to compare RHO staining in the region where recombination of the M2 exon has not occurred (lack of PKM1 staining) with the regions where the recombination has occurred (appearance of PKM1 staining in the rods). At this time point, one can appreciate a small decrease in RHO^+^ outer segments in the recombined region and a slightly longer outer segments of the rods that do not show PKM1 expression in the same retina. We also wanted to assess if PKM2 deletion makes the photoreceptors vulnerable and reduces their survival over the period of time. The data from 8-month-old animals is included in the revised draft. We also examined aged mice (74-85 weeks old) for surviving photoreceptors by plotting number of rows of photoreceptor nuclei from the center of the retina (optic nerve head) to periphery for the typical spider plots. We did not observe an apparent difference between PKM2^fl/fl^; Rod-cre retinae and Bl6/J age-matched controls. Due to the mosaicism of the Rod-cre line used in our experiments, we may not have been able to detect a phenotype that perhaps would be more obvious in the full ko. Increasing the number of aged animals or using alternative photoreceptor Cre lines are approaches that we are currently undertaking. Using Cre lines that exhibit less mosaicism (than the currently used Rod-cre mouse line) might have an added advantage of giving more penetrant metabolic phenotypes with LDHA or PKM2 conditional animals. Since these results would not change our conclusions about the role of LDHA and PKM2 in the glycolytic preference of photoreceptors, we are not including the results of those experiments in the interest of the amount of time it would take to breed and then age such animals.

*14) In the subsection “Fibroblast growth factor signaling regulates anabolism”, the authors should cite a recent paper, PMC5121888, that also confirms that Y105 of PKM2 is phosphorylated in mouse retinas.*

The work by Rajala et al. (2016) is now cited. Their conclusions derive from detection of the signal using IHC and western blot using the phospho-Y105-specific antibody. They could detect the signal in photoreceptors on IHC (that predominantly express PKM2) as well as in the IPL, as well as in what appears to be Mueller glia (all of which would arise from PKM1 and not PKM2). They also see an increase in signal in all of these locations in a light and activity-dependent manner. Of critical importance to such an approach is the establishment of the phospho-specificity of the antibody, and ruling out an apparent increase in signal intensity due to an increase in the total protein itself. To their credit, they have been able to observe this phosphorylation on IHC and we hope that the issue of phospho-specificity was addressed during peer-review, though we did not come across this aspect in the paper itself. If the phospho-specificity of the anti-Y105 antibody on IHC is indeed established according to the authors’ own data it would indicate PKM1 is regulated in a similar manner as PKM2. On the other hand, the caveat with using whole retinal lysates is that a likely scenario of PKM1 phosphorylation at Y105 cannot be ruled out as a confounding factor (as we highlight in Figure 4 and which is possible as shown by phosphorylation of PKM1 in muscle lysates in 4B).

However, our results align in many respects and we highlight both of these aspects in the text (subsection “Fibroblast growth factor signaling regulates anabolism”, first paragraph).

*15) Please include MW marker positions on Figure 4. Also, please explain what the "isotype-matched" means.*

The marker positions are now included. The concern of the reviewer regards the apparent molecular weight similarity of PKM2 (~56 kDa) with IgG heavy chain. Our method of immunoblotting (following immunoprecipitation) using a conformation-specific secondary antibody eliminates a likelihood of obfuscating bands due to denatured immunoglobulin chains.

An isotype-matched rabbit monoclonal (IgG) was used as a control for non-specific binding in IP experiments using PKM2 rabbit monoclonal. This is now explained in the figure legend.

*16) The authors should be careful to not over-interpret data in some of their descriptions. For example, the authors state "Impact of glycolytic perturbation on nucleotide availability was directly visualized…". Since FGF signaling can influence other processes besides glycolysis I think this is over-interpreting the data. It would be better to just say that FGF receptor signaling influences nucleotide availability and this could be linked to limitation of glycolysis. Alternatively the connection could be strengthened by evaluating the effects of PKM2 or LDHA inhibition on EU incorporation.*

The advantage of the EU incorporation assay was that it enabled us to directly visualize nucleotide incorporation in nascent RNA in photoreceptors. In our opinion this assay is much better than other metabolite assays where whole retinal lysates are needed. We stand by our claim but have modified the sentence to incorporate the reviewer’s suggestion. We do show the effect of Ldha inhibition by oxamate. We agree with the reviewer that FGF inhibition has a much stronger effect than oxamate itself, which is likely due to the role that protein tyrosine phosphorylation plays at multiple nodes (LDHA, PKM2 and Pyruvate dehydrogenase kinase) in regulating the glycolytic metabolism in addition to other nonglycolytic roles that FGF signaling might have. We have tried multiple effectors of PKM2-specific activity (DASA, TEPP and ‘Compound 9’). All of these have been shown to significantly affect aerobic glycolysis in cancer. But in our hands these agents did not change the lactate production from the retina, the quaternary state of retinal PKM2, or nucleotide incorporation in photoreceptors. We are not certain if the differences are due to the efficacy of these drugs (though we used three different drugs at a range of concentrations) or some innate differences in PKM2 protein milieu or its regulation in photoreceptors and cancer cells that may underlie differences in retinal and cancer aerobic glycolysis.

*Reviewer #2:*

*Vertebrate photoreceptors are among the most metabolically active cells, exhibiting a high rate of ATP consumption. This is coupled with a high anabolic demand, necessitated by the diurnal turnover of a specialized membrane-rich organelle, the outer segment, which is the primary site of phototransduction. It is not clear to date that how photoreceptors balance their catabolic and anabolic demands. The current study has attempted to address this fundamental issue in photoreceptor biology. The authors have used several biochemical, immunological, genetic and viral transduction methods to address the importance of glycolysis on the outer segment biogenesis. The study is interesting but failed to support the authors claim that glycolysis regulates the outer segment biogenesis.*

The major criticism of the reviewer is “The study is interesting but failed to support the authors claim that glycolysis regulates the outer segment biogenesis.” Our conclusion from the data differs from the referee’s summary of our work for primarily two reasons:

1) Our approach of uncovering reliance on glycolysis utilized functional perturbations and assessments of three different nodes of glycolysis, establishing genetic necessities for key players, metabolic nonequivalencies of glycolytic enzyme pairs (Ldha vs Ldhb and PKM1 vs PKM2) and genetic dissection of allosteric control of photoreceptor glycolysis by metabolic complementation. We have consistently observed dependence of rod outer segment length on a functional glycolytic pathway.

2) “[…]glycolysis regulates the outer segment biogenesis.”: We have refrained from a general statement that would implicate a metabolic pathway of broad survival significance as a controlling mechanism in biogenesis of a specialized organelle. On the contrary, we have posited that the multiple metabolic demands posed by outer segment shedding and a need to replenish it, require a clever approach on part of cells. The summary of our findings is that photoreceptors rely on aerobic glycolysis which they themselves carry out and that this pathway is subject to regulatory strategies at multiple levels. We show that perturbing these regulatory steps results in visibly shorter outer segments and reduced steady state concentration of metabolites that are necessary for anabolism. We have not concluded that glycolytic pathway can regulate the complex aspects of OS biogenesis, but instead infer that photoreceptors rely on a functional glycolytic pathway to enable OS growth.

*1) In these studies, authors have examined the isoform expression of LDH, PKM2, then phosphorylation state of PKM2 under dark- and light-adapted conditions, identified FGF signaling promotes PKM2 phosphorylation and splicing regulation of PKM1 and PKM2. Some of these studies have been done before by other labs (see below), which the authors did not acknowledge or reference in their manuscript. These include:*

*• Identification of LDH isoforms (Casson RJ et al. 2016 IOVS).*

*• PKM1 and PKM2 isoform characterization, light-dependent tyrosine phosphorylation of PKM2 (Rajala et al. 2016 Sci Rep).*

*• The authors claimed in this study that fibroblast growth factor (FGF) signaling was found to regulate glycolysis through phosphorylation of PKM2. This finding is not novel as it has been elegantly shown in tumor cells that FGFR regulates PKM2 phosphorylation (Hitosugi et al. 2009 Sci Signal).*

*• Regulation of PKM1 and PKM2 splicing (Su et al. (2017 Mol Cell Biol)*

*The authors are urged to cite these earlier references and give proper credit for these studies. They could discuss how the published results are similar or differ from their observations in this manuscript.*

We are addressing the 4 specific issues raised by the reviewer here in comment#1 itself, and not later if they appear redundantly in comments 2 through 16.

“Identification of LDH isoforms (Casson RJ et al. 2016 IOVS).”: Casson et al. have not identified LDH isoforms in the retina. They have reported Ldha expression in the retina by western blot and IHC in multiple species. However, missing from their analysis and critical for the interpretation of significance of their work in terms of glycolysis, is the expression pattern of Ldhb. This is especially important because before them Michael Kalloniatis’ group has reported that, in the retina, Ldha and Ldhb subunits can assemble in all 5 possible combinations (Acosta et al., 2005. PMID: 16163270) underscoring the importance of careful analyses of all isoforms in cells of interest. Secondly, we do not endorse the assumption that mere expression of enzymes implicated in cancer signify a similar anabolic dependency. The PK isoforms studied by Casson et al. and Ldha are known to be expressed in many other cell types and not just in cancer cells. For example, skeletal muscle expresses Ldha, but does not carry out aerobic glycolysis. On the other hand, Ldhb can promote a Warburg-like phenotype as well (Oginuma et al., 2017. PMID: 28245921). Their work provides a descriptive catalog of expression of PKM1, PKM2 and Ldha in many mammals and we acknowledge that. But we disagree with the reviewer’s specific assessment of Casson et al.’s work.

“PKM1 and PKM2 isoform characterization, light-dependent tyrosine phosphorylation of PKM2 (Rajala et al. 2016 Sci Rep).”: We had not come across Rajala et al.’s and Rueda et al.’s work when we were preparing our manuscript. It was inadvertent omission on our part. We are now citing these papers. Though already addressed as an issue in response to reviewer#1, we would like to reiterate that although we are glad to find similarities in the manner in which tyrosine kinase signaling is regulating PKM2, Rajala et al.’s results should be considered as PKM regulation as opposed to PKM2. We are discussing it in more detail in the text.

“The authors claimed in this study that fibroblast growth factor (FGF) signaling was found to regulate glycolysis through phosphorylation of PKM2. This finding is not novel as it has been elegantly shown in tumor cells that FGFR regulates PKM2 phosphorylation (Hitosugi et al. 2009 Sci Signal).”: Our interpretation of work by Hitosugi et al. differs from that of reviewer’s. Hitosugi et al. demonstrated direct phosphorylation of PKM2 by oncogenic forms of FGF receptor 1. They model a translocation event where the C-terminal kinase domain of the FGF receptor is fused with a self-association motif of ZNF198, rendering the kinase constitutively active. The cells stably expressing this fusion protein lose their dependence on interleukin 3, which is observed in the parent cell line. Thus, it is not FGF signaling per se but direct phosphorylation by the kinase domain of *FGFR1* that is responsible for PKM2 phosphorylation. The authors show direct binding and phosphorylation of PKM2 by *FGFR1*. These authors have also shown constitutively active versions of other tyrosine kinases can also result in PKM2 phosphorylation at Y105. This varies significantly from the reviewer’s interpretation. Secondly, our intention was not to discover a novel aspect of PKM2 regulation by tyrosine kinase signaling. We wanted to know the specific pathway that regulates PKM2 tyrosine phosphorylation in the retina. To our knowledge, this signaling pathway was not known to interact with PKM2 in the retina. Thirdly, it is not only PKM2 that is regulated by FGF signaling. As Figure 4 shows, Ldha is also a target of this pathway. Subsequent panels in Figure 4 aim to demonstrate the role of FGF signaling in regulating aerobic glycolysis in the retina, which we believe are exciting data and worth exploring further.

“Regulation of PKM1 and PKM2 splicing (Su et al. (2017 Mol Cell Biol)” and comment (14) from below “Figure 3—figure supplement 2 does not add any new information. The authors' data show opposite expression of these splicing factors. There was a study recently published showing that RBM4 Regulates Neuronal Differentiation of Mesenchymal Stem Cells by Modulating Alternative Splicing of Pyruvate Kinase M (Mol Cell Biol 2017).”: We believe that these comments represent drastically different interpretation of our findings and those of the existing literature. To our knowledge the molecular basis of biased expression of PKM1 and PKM2 isoforms in the two different layers of the fully differentiated retina is not known. To us the simplest explanation was that it could be attributed to splicing factor expression, which in turn would dictate the bias and explain how postmitotic photoreceptors keep expressing PKM2. While expression of *SRSF3* aligns with the existing model for preferential inclusion of M2-specific exon, we expected *PTBP1* expression to follow a similar pattern, but it was not. Thus the reviewer’s implication that somehow the mutually “opposite” expression of these splicing factors is expected goes against how we interpret the literature. Secondly the reference provided by the reviewer (Su et al., 2017) (that was published a week before our submission) looks at RBM4 splicing factor-dependent PKM1 expression by antagonizing Ptbp1. This reference still does not explain our observation that Ptbp1 is enriched in the cells that preferentially express PKM1 in the retina. The RBM4 paper would predict that Ptbp1 splicing activity in the INL would be attenuated, in order to promote PKM1 expression. We still remain puzzled by the observation that PTBP1 is expressed at higher levels in the INL. A relevant reference for retinal splicing that touches on PTBP1 is the work by Murphy et al., (PMID: 2754135), although their work still does not explain how PKM2 is expressed in photoreceptors. Again, the splicing choice of photoreceptors is peripheral to our primary focus on glycolysis, but we thought of highlighting the discrepancy between existing splicing models and our observation by providing these data on the expression of the splicing factors.

*2) LDH isoforms identification has previously been reported (Casson RJ et al. 2016 IOVS). The authors must cite this manuscript.*

Addressed above.

*3) The authors stated in the manuscript (subsection “Lactate producing isoform of Ldh in photoreceptors”, last paragraph) the recombination efficiency with rod-cre varied between 50-90%, but that is not correct. The rod-cre used in these studies will not recombine more than 50%. The authors have shown only protein expression by Western blots. They need to show the deletion by immunohistochemistry.*

We show that the cre line is mosaic and in the sample section shown in Figure-1, figure supplement 2, there are more than 50% rods that show recombination history. Is the reviewer stating that the recombination was maxed out at 50% from his/her experience? It is worth noting that expression of engineered alleles in mice can be variable even in inbred strains.

*4) The authors have shown that lactate production was significantly reduced in conditional LDH-A mice. Why did the authors not study OS biogenesis in LDH-A deleted mice? The shRNA strategy is not well justified. Have the authors examined the OS in LDH-A KO mice? Generally, the shRNA approach may not knock-down completely the gene of interest but conditional deletion will? The authors observed the shortening of OS. Could this be an off-target effect? LDH-A is also expressed in other layers of the retina (INL and IPL).*

*Immunohistochemistry is not the ideal way to demonstrate OS length; hence the authors measured IS+OS. The authors should use ultrastructural studies, such as EM or high resolution LM to demonstrate the OS length phenotype. Some of the micrographs show thinning of the outer nuclear layer thickness (e.g., DAPI stained sections in Figure 1, Figure 2, and 3C, suggesting retinal degeneration in these genetically modified retinas, which could argue against shortening of OS length. Did the authors do TUNEL or any other test for dying photoreceptor cells?*

Though we have already stated our reason to use electroporation and not rely on knockout for cellular phenotypes, we are stating them here as a response. Our primary concern was to establish the cell autonomous requirement of aerobic glycolysis for the photoreceptors and hence we chose an electroporation-based approach. Our major concern with knockouts affecting large numbers of cells within a specific area was that it might change the metabolic environment that the photoreceptors sit in. This would cloud our interpretation of the autonomy of a photoreceptor phenotype, as it might reflect indirect effects on Mueller glia and/or RPE, etc. Another possible problem with a large swath of cells knocked out is that, if we did not observe a phenotype, it could be due to a cellular adaptation to an altered environment. This concern has recently been highlighted by the work of Martin Friedlander (Kurihara et al., 2012. PMID: 23093773) who show that in the face of catastrophic metabolic alterations brought about by ablation of the choriocapillaris, the rods are able to persist for months and maintain a functional response, indicating remarkable adaptability. Also a gene knockout might not necessarily recapitulate the phenotype observed with acute interference such as RNAi (e.g. *Doublecortin* and TUG1 give phenotypes with sh knock down but not in full KOs). Though the concerns usually stem from functional compensation in case of whole body knockouts, one cannot effectively rule out a similar effect in conditional knockouts as well. Additionally, electroporation enables us to address issues that cannot be addressed via classical conditional alleles, such as establishing that catalytic activity of Ldha is essential (rather than the entire protein itself), allowing us to devise an experimental strategy to target an allosteric metabolite (F-2,6-BP) which would not be otherwise straightforward due to three genes (*PFKFB1, PFKFB2* and *PFKFB4*) (issue of functional redundancy and ruling out pleiotropic effects) expressed in photoreceptors, non-equivalent roles of PKM1 and 2, and Ldha and b.

Regarding off-target effects of shRNA, we showed that we could complement the shRNA phenotype by a non-targetable construct of the targeted gene. This is the gold standard control for specificity.

We question the usefulness of EM ultrastructural studies for the primary focus of our study. First, it would be difficult to pinpoint the affected photoreceptor outer segment amidst other normal looking OSs. Secondly, our method of coelectroporation with mGFP allows one to reliably assay the shortening of outer segments, which we believe is sufficient for assessing the effect of glycolysis. Though there are advantages offered by EM studies, these are tangential to our goals of establishing the function of a metabolic pathway for photoreceptor cells. However, such studies would be well suited for understanding the cellular responses at organellar/suborganellar level, which we would very much like to know.

Thinning of the ONL is also observed in dark-reared sh transfected retinae, but there is a rescue in terms of OS length. Thus retinal degeneration cannot be a causative factor for OS shortening. In addition, the non-electroporated neighboring rods have outer segments that are identical to those from rods outside the electroporated patch. We have addressed this issue in the Discussion section. These data indicate that degeneration is not responsible.

We have done TUNEL staining of these retinae. We did not observe TUNEL positive nuclei at the timepoints shown in the manuscript. However, we cannot rule out the occurrence of cell death since we do not know how long a dead cell persists in the adult retina in a state capable of being captured by TUNEL. Given that the cells would die asynchronously, and there was not a large number of cells missing, we estimate our chances of detecting a TUNEL-positive nucleus in a given retinal section to be very low, even if there was death.

*5) Could shRNA knock down in other retinal layers (may be Muller cells) may indirectly affect the structure of OS?*

We have included rhodopsin staining of nonelectroporated cells. In the case of non-autonomous effects, these cells would have been expected to have been affected, but this was not the case. We discuss this issue in the Discussion. Also to note is the experiment with sparsely electroporated PKM2sh retina, which further reduces the possibility of non-autonomous effects.

*6) Figure 1 – There is no evidence of disc shedding in this experiment. Could there be less opsin trafficking to the OS? Such a possibility cannot be ruled out. This experiment is overstated.*

We do not claim it to be disc shedding (“[…]presumably due to less disc shedding). We have incorporated the suggestion of reference for this effect made by another reviewer. Our understanding based on the literature is that opsin trafficking is necessary for OS biogenesis, and thus we did not consider less opsin trafficking to be a realistic event that could rescue the photoreceptors in dark.

*7) The statement "lactate production by the photoreceptors cannot be attributed to lack of mitochondrial activity." The authors have done experiments on (Figure 1) wild- type retinas. Have they carried out these experiments in LDH-A-knockdown or KO mice?*

We have now included cytochrome oxidase assay on *LDHA* KO mice (Figure 1—figure supplement 4). We do not observe a decrease in COX activity after *LDHA* deletion.

We would like to reiterate that the purpose of assaying for mitochondrial activity in wild type retina was to examine if a cause-and-effect relationship exists between *LDHA* expression and mitochondrial function. We observed that despite having normal functioning mitochondria, photoreceptors have *LDHA* expression. Thus, LDHA-dependent aerobic glycolysis seems to be a choice as opposed to a consequence of poor mitochondrial activity. Thus, photoreceptor lactate production fits the definition of aerobic glycolysis.

*8) Figure 2 – what is the rationale to regulate TIGAR expression spatially and temporarily? In Figure 2 the authors used AAV-mediated expression of TIGAR. Lactate levels were done in AAV-TIGER (Figure 2) but not for Figure 2? It is very confusing, and there was no rationale provided for these experiments. It seems that authors may have difficulty in measuring lactate levels for the inducible expression system?*

The rationale is already stated in the text (subsection “Allosteric regulation of glycolysis in photoreceptors”, last paragraph). We understand the reviewer’s concerns that the multiple reasons for using TIGAR overexpression could be confusing. We have modified our description of this strategy in an effort to improve the clarity of our rationale.

*9) Figure 3 – PKM1 and PKM2 expression has recently been reported (Rajala 2016 Sci Rep). The authors have not cited this reference.*

As stated before, this was an oversight on our part. We did cite work by Lindsay et al. (2014) who first characterized these isoforms in the retina, but not every subsequent work that follows theirs’ unless the data differed from ours or the work provided insights that could not be obtained from Lindsay et al. Though, to our knowledge, the first evidence of PKM2 in the retina is attributable to the work of Santa Ono and colleagues (Morohoshi et al. 2012. PMID: 22465421).

*10) The authors have shown the developmental expression of PKM2 and PKM1 on western blots (Figure 3), which is not the ideal way to show the developmental expression. If the authors wanted to show this, they should provide immunohistochemistry or ISH.*

The developmental milestones at indicated time points in the postnatal retina are already well established. Since the increase in PKM1 expression is very gradual and we did not observe a remarkable change in expression level at any of the time points, we did not think it warranted further in-depth analysis. This manuscript is focused on the adult retina and we included the time course data simply out of interest.

*11) The authors used rod-cre to delete PKM2 and measured LDH activity (Figure 3). For structural studies, they used shRNA and examined OS length (Figure 3). These studies are not convincing. Why did they not observe similar shRNA effects with conditional PKM2 KO mice?*

We would like to clarify that we did not measure LDH activity in these retinae as the reviewer suggests. The rationale for using shRNA is already described previously. We do observe a slight reduction in outer segment length in PKM2 conditional retinae very early. With progressive age the OS length and the condition of the retina worsens.

*12) There is no indication of how much PKM2 is deleted or knocked down. In the absence of these experiments, it is very difficult to interpret the data. Moreover, the authors did not carry out any functional studies, such as ERG to examine the role of PKM2 in photoreceptor functions?*

The knockdown of PKM2 and steady state protein levels were already depicted in Figure 3—figure supplement 1 (Figure 3—figure supplement 2 in the revised manuscript).

*13) Figure 3—figure supplement 2 does not add any new information. The authors' data show opposite expression of these splicing factors. There was a study recently published showing that RBM4 Regulates Neuronal Differentiation of Mesenchymal Stem Cells by Modulating Alternative Splicing of Pyruvate Kinase M (Mol Cell Biol 2017).*

This comment is already addressed before.

*14) The authors stated that PKM2 deletion upregulates PKM1, but has no effect on photoreceptor structure (Figure 3—figure supplement 1). On the other hand, forceful expression of PKM1 had a reduction in the length of OS? How do authors explain this discrepancy?*

In rods where recombination occurred, and expression of PKM1 resulted, slightly smaller outer segments occurred, compared to non-recombined neighbors in the same retina. The initial characterization of the PKM2 conditional allele (Israelsen et al., 2013) has already shown that deletion of the M2-specific exon results in appearance of PKM1, but the expression levels are rather low (~40% of the PKM transcript are PKM1 and PKM protein steady state suggests lower expression level than the non-recombined cells). Thus, we speculate that the PKM1 expression levels from the two genomic loci in the conditional knockout never match the higher levels achieved by PKM1 overexpression driven from multiple copies of plasmid using the strong CAG promoter.

15) Figure 4 – FGF signaling – Authors have identified that FGF signaling promotes the phosphorylation of PKM2. It is not a novel finding. It has been shown in tumor cells (Hitosugi et al., 2009). The authors have not acknowledged this information in the current manuscript.

This comment has been addressed before.

In rods where recombination occurred, and expression of PKM1 resulted, slightly smaller outer segments occurred, compared to non-recombined neighbors in the same retina. The initial characterization of the PKM2 conditional allele (Israelsen et al., 2013) has already shown that deletion of the M2-specific exon results in appearance of PKM1, but the expression levels are rather low (~40% of the PKM transcript are PKM1 and PKM protein steady state suggests lower expression level than the non-recombined cells). Thus, we speculate that the PKM1 expression levels from the two genomic loci in the conditional knockout never match the higher levels achieved by PKM1 overexpression driven from multiple copies of plasmid using the strong CAG promoter.

*16) Figure 4 – PKM2 undergoes a light-dependent tyrosine phosphorylation on Tyr105. These studies have recently been reported by Rajala et al. 2016 (Sci Rep). The authors have not acknowledged this study.*

We have addressed this comment above.

*Reviewer #3:*

*This manuscript provides novel and interesting data on the reliance of aerobic glycolysis for photoreceptor outer segment renewal. Overall, the paper is very good and a significant contribution. However, there are some significant problems that need addressing before the results and conclusions that are presented can be accepted. In addition, there are several additional items that are off putting and overstepping the presentation and results.*

*First, the authors are only addressing aerobic glycolysis in the rod inner segments and outer segments. There does not appear to be any data for cones or rod and cone photoreceptor synapses. So, the title should reflect this and state something akin to "Aerobic glycolytic reliance promotes anabolism in rod photoreceptor outer and inner segments."*

Our work does not address aerobic glycolysis in a given cellular compartment, but uses the readout of outer segment length as an assay of perturbations of aerobic glycolysis. The perturbations might affect other cellular compartments, but our assay has been only of outer segment length, except in the case of acute FGF inhibition, where we saw an effect on nascent RNA synthesis in the nucleus. But given that this work highlights the propensity to preferentially carry out glycolysis (glycolytic reliance) in order to maintain outer segments, nucleotides and NADPH (anabolism) in photoreceptors, we believe that the current title is accurate. Further studies might uncover if the metabolic effects of specific perturbations vary among specific compartments.

*Second, the authors should provide more detailed and important information in the Methods. This information is critical for proper interpretation of the results. For example, what the light level in the animal room and cages, what time of day were the mice sacrificed, what area and quadrant of the retina was quantified for OS-IS measurements, how was this location established, what are the measurements for OS and IS alone as they could both change, how many non-adjacent sections per retina were quantified per mouse, what was the volume of incubation for the lactate assay, what type of confocal was used and how many Z-stacks were there per image, what was the exact age of mice used for the histology as in the subsection “Dissections and adult explant cultures”, the authors stated that they used P23-P28 mice. According to LaVail 1973 J Cell Biol, the outer segments in the C57BL/6J mice reach their rate of synthesis and disposal at around P21-25.*

- Light level: This information is now provided

- Time of the day of harvest: Most of the animals were euthanized for tissue harvest between 3-9 hours after lights were turned on. Specific time points are reported for experiments of dark-reared and light-reared animals and AAV-infected retinae for lactate assay (where we wanted to avoid small differences due to possible circadian/diurnal changes). Whenever possible, animals belonging to the same assay group were sacrificed together. All data points are presented, i.e. we do not exclude any outliers.

- Quadrant of the retina: Electroporation of the retina yields a patch of transfected cells. The exact shape and size of the patch varies among animals, though roughly in the same place. The extent to which the DNA will spread in the subretinal space and how many cells will get transfected cannot be predicted. In the experiments presented here, the injections targeted the dorso-nasal quadrant of the right eye, as a matter of experimenter preference, though in some cases the transfected cells may be from a nearby area.

- OS and IS measurements: In an ideal situation we would have quantified these two compartments separately. However, we resorted to IS+OS quantification in two possible situations:

i) In cases where a clear boundary between IS and OS was lost

ii) In cases where the OS was present but too small to accurately measure by itself

- Number of nonadjacent sections: For each electroporated retina that was sectioned on multiple slides, we chose the section that had the biggest electroporated patch. We aimed to quantify all the photoreceptors in this patch, which involved multiple nonoverlapping visual fields, by moving along the x-axis in the electroporated patch. This approach samples across a large area and thus can be considered equivalent to a nonadjacent section which would sample along the z-axis rather than along the x-axis.

- Type of confocal and number of Z-sections: Confocal type is already provided in the Materials and methods subsection “Immunohistochemistry”. There was a typo in the model number (LSM710 was mentioned as LSM10). It has been corrected. This information now also includes another confocal that was used for some experiments in the revision. The number of Z-sections varies between individual sections and is governed by the number of electroporated cells and their location in the tissue slice. For some sections we could sample all the photoreceptors in 12 slices, while for others we did as high as 20 slices.

- Exact age of mice used in subsection “Dissections and adult explant cultures”: First we would like to clarify that we have used these mice for explant culture and not histology for outer segments. Though we are not certain of the reviewer’s concern that the choice of age-range would pose for metabolic assays, we speculate that he/she is concerned about changes in the metabolic capacity of the photoreceptors. As stated by the reviewer, most photoreceptors achieve the adult synthesis and shedding rates by after 21 and 25 days. According to those data, between 23 and 25 days the mean OS length increases by 10% in length with ~70% overlap in the range of data. We have quantified lactate secretion from fresh untreated retina many times (from what would essentially cover the entire age range mentioned in the manuscript) in multiple different experiments. We went back and examined if lactate secretion was lower in experiments when we used 23-25 day old mice. There does not seem to be a trend or significant difference to indicate that a difference of two/three days in that age group would make in glycolytic capacity. In addition, the siblings from the litter (or litters of the same age) were used in an experiment that also included controls. Thus we believe that age differences for any data points that come from mice 23 or 24 days old should not contribute to a statistical differences.

- Volume for lactate assays: 0.5 mL. Can be found in the Materials and methods subsection “Dissections and adult explant cultures”.

*Third, the authors conclude from looking at their retinas 42-45 days following electroporation that the shortened rods were due to decreased synthesis due to decreased aerobic glycolysis. This cannot be accepted at face value without conducting two additional experiments: show rod disc synthesis results [e.g., per Young and Bok studies] and rod phagocytosis results [e.g., per LaVail studies].*

We conclude that OS length is dependent on functional glycolytic pathway. By three separate genetic perturbations targeting the glycolytic pathway at three different nodes, we consistently observe that the OS length reduces. The work by Richard Young and Matthew LaVail has indeed provided experimental evidence that has laid the groundwork for studies on OS biogenesis. The work utilizes EM ultrastructural studies. While those assays give unbeatable resolution, they are either not compatible or incredibly difficult to do when merged with our approach of electroporation and isolated cell labeling. Even if we are able to successfully implement those methods with our assays, the information on the dynamics of OS biogenesis after glycolytic perturbation can be inferred but the final OS length (which is an important end result) will still need to be assessed. For the latter part, confocal microscopy is reliable and sufficient.

*Fourth, please explain why eye opening occurred on day 11 in these studies. Most published mouse paper find that this event occurs at postnatal day 14 +/- 1 day.*

We followed our mice every day to assess eye opening so as to ensure that the animals were not exposed to unwanted amounts of light. We were concerned that the eye of the electroporated animal might open earlier than usual because of the incision at P0 made to expose the eye for subretinal injections. We found the eyes opened ~postnatal day 11, whether they were injected or not. These mice were transferred to dark, as reported. Figure 5 shows an injected animal at P11 and an uninjected animal at P12. As can be seen in both the images, the eyes are open. To our knowledge, the beginning of eye opening is also used as guide for staging mice at P11 (Please refer the JAX mouse staging chart at https://oacu.oir.nih.gov/sites/default/files/uploads/training-resources/jaxpupsposter.pdf)

Author response image 1.**DOI:**
http://dx.doi.org/10.7554/eLife.25946.028

*Fifth, the choice of words in several places throughout the manuscript are overstated or not properly used. This often results from lack of citation of previously conducted work. For example, they state that "The cell types that carry out aerobic glycolysis in the normal adult retina have not been determined." This is just false. The work of Rueda et al. published in Mol Vis in 2016 that shows cell-type glycolysis, aerobic glycolysis, high energy transferring kinases and oxidative phosphorylation in over 20 different compartments and cells in the retina. A summary figure clearly shows all of this data. However, the authors have not cited this manuscript anywhere in their manuscript.*

*For example, they state that "surprisingly the steady state levels of ATP…did not differ from controls." Winkler in J Gen Physiol 1981 showed that retinal ATP levels are steady under most conditions: not surprisingly since Lowry and co-workers demonstrated in the 1960s that this energy measure is the last to change even after anoxia.*

Rueda et al. looked at expression of enzymes participating in intermediary metabolism. Due to the exhaustive and descriptive nature of their study, understandably there is no functional assessment of their participation in aerobic glycolysis. One can speculate about the propensities of a cell type based on expression analysis and the model does just that. Their work is a comprehensive analysis of expression and has enormous descriptive value, which will greatly benefit the field. The glycolytic pathway has seen much investigation recently, thanks to the impetus from its relevance to cancer. Based on these developments, we believe it is the right time to investigate the functional significance of aerobic glycolysis as a phenomenon and differentiate it from housekeeping glycolysis. This type of functional evaluation is required, as one cannot conclude functions from expression of glycolytic genes alone. Thus we stand by our conclusion that the cell types responsible for aerobic glycolysis are not known since we are unaware of any functional assessment of this phenomenon (not to be confused with role of glycolysis in retinal physiology).

It is important to refer the context in which we have used the statement on line 90 of earlier draft. We had observed a significant decrease in lactate production upon oxamate treatment. We anticipated a decrease in ATP levels after glycolysis inhibition. However this was not the case. The ATP level did not change despite altered lactate levels. Reducing lactate levels would have slowed glycolysis. However, slowing glycolysis did not change ATP- this was surprising. Secondly, Barry Winkler’s work (that includes the reference cited by the reviewer) does indicate that ATP levels can change in acutely treated retinae, depending on the context. For example, 10 mM KCN causes a massive and rapid drop in ATP levels (Winkler et al., 1997. PMID: 9224285). Another point to note is that our oxamate treatment occurs for hours and it cannot be considered acute, using Winkler’s definition. Finally, Barry Winkler’s work (Winkler 1981, cited by the reviewer) shows that inhibition of glycolysis by iodoacetate results in an ATP decrease. Also, comparing the effects achieved by inhibiting housekeeping glycolysis (Winkler’s IAA) with aerobic glycolysis (LDH, oxamate), one can consider the effects as surprising.

*For example, in the subsection “Aerobic glycolysis in the retina”, the authors incorrectly stated that adenylate kinase (AK) synthesizes ATP. The do not acknowledge that mouse retina also expresses creatine kinase (CK). The authors need to state that CK and AK serve to regenerate the ATP by reversible reactions that respond to the law of mass action. The implication of these enzyme reactions need to be explained as these enzymes may respond under conditions of high ATP hydrolysis or raising [ADP] (E.g., see the work of Wallimann, Linton et al. 2010; Rueda et al., 2016).*

We are including mention of CK along with AK as phosphotransfer enzyme systems that can maintain ATP steady states. However, this does not change the conclusion of the experiment with azide that indicates that after aerobic glycolysis attenuation, the retina is more reliant on the mitochondrial contribution for ATP level maintenance.

*For example, the authors state that "presumably as a result of less shedding". This results from the process of "photostasis' as published in Exp Eye Res 1n 1986 by Penn and Williams.*

We appreciate pointing to us the reference. We are amending the sentence to encompass the regulatory mechanisms.

*For example, the authors state that "Our work expands the cell types where aerobic glycolysis can occur to include a mature cell type, the differentiated photoreceptor cell". As noted, the authors did not cite the prior work of Rueda et al. published in Mol Vis in 2016. This work also pyruvate competition in LDH histochemistry assays and confocal to demonstrate that Ldh-A was preferentially located in photoreceptors[…]not cited in the first paragraph of the subsection “Lactate producing isoform of Ldh in photoreceptors”, as confirmatory. So the current manuscript "confirms and definitely expands" would be the correct terminology.*

We agree that Rueda et al.’s findings should have been cited where parallels exist or where they do not agree with ours. As mentioned before, it was an oversight on our part while we were preparing our manuscript. We have since made changes to cite that work. Though our findings on expression of key aerobic glycolysis players agree, our interpretation of that paper is that the authors did not demonstrate aerobic glycolysis in the photoreceptors or the importance of that process. Demonstration of Warburg effect, by definition, critically requires production of lactic acid in oxygenated condition and in presence of functional mitochondria. Rueda et al. demonstrate photoreceptor-enriched expression of Ldha. However, Ldha is also expressed in skeletal muscles, which do not carry out aerobic glycolysis (lactic acid is produced when the need for ATP generation cannot be met by supplied oxygen). Also, it is of critical importance to consider the properties of Ldh isoforms in the context of IHC used by Rueda et al. An all-Ldha tetramer is equally adept at generating lactate as well as carrying the reverse reaction i.e. oxidizing lactate to pyruvate. Clearly the direction of the reaction in a cell is determined by many factors including, but not restricted to, the rate of pyruvate generation and ratio of NADH:NAD+ and cannot be predicted by IHC alone. So, does expression of Ldha in photoreceptors necessarily mean the cell carries out aerobic glycolysis? Why would Ldha not participate in lactate utilization, as per Mueller-photoreceptor lactate shuttle? Also, they cannot exclude the absence of Ldhb in the photoreceptors. They use LDHA-specific antibody in conjunction with the pan-Ldh antibody, which supposedly detects both Ldha and Ldhb. The subcellular staining pattern, especially in the photoreceptors, is markedly different when the two antibodies are compared.

Lastly, the ldh assay cited by the reviewer needs to be considered fully. All the Ldh isoforms can oxidize lactate to pyruvate and generate reducing equivalents for the colorimetric NBT-dependent reaction. As mentioned above, even the LdhA-only isoform (aka ldh5) can catalyze the reaction with equal ease in both the directions and can generate the reducing equivalents when lactate is given as a sole substrate. Inhibition by pyruvate is a great way to ascertain Ldhb function in vitro. But as pyruvate concentration is increased during the histochemical reaction, wouldn't the Ldh strive to establish equilibrium and generate less and less of reducing equivalents in that process? Our guess is that the decrease in staining intensity in the IS (Figure 8H of Rueda et al.) might reflect that process and not necessarily inhibition by pyruvate. How does the staining look when the molar ratios for lactate and pyruvate are flipped? Does an absence of staining in that case suggest inhibition of enzyme activity (as might be the case for Ldhb) or fewer reducing equivalents generated to make the blue ppt due to more pyruvate to lactate conversion (which Ldha might still do)? In addition, the enzyme activity assay on retinal section cannot be considered equivalent of lactate production by the retina. Thus, while we do agree that our Ldha expression patterns match, we cannot attribute functional assessment of aerobic glycolysis to their work.

*For example, the Abstract says that the process of photoreceptor catabolism and anabolism is "poorly understood". It should be more precise as there are many studies on these process, but not in the context of rod outer segment biosynthesis.*

Though there have been ongoing studies on retinal metabolism with an underlying motivation to understand photoreceptor physiology, our understanding of how central carbon metabolism is regulated in these cells significantly lags behind that of other cell types, like pancreatic β cells, hepatocytes or even dendritic and T-cells of the immune system. Understandably, working with retinal tissue, especially for metabolic assays, poses unique challenges and we fully appreciate those and the efforts of researchers in this field. In this regard, the Abstract specifically mentions, “How photoreceptors balance their catabolic and anabolic demands is poorly understood”. Given that many genetic tools needed to probe glucose metabolism have become available within last 5 years, the question has not been adequately probed and our work intended to do just that.

*Sixth, no retinal references are provided for COX and SDH histochemistry. Several previous papers conducted these experiments in rodents (SDH in rat: Hansson, Exp Eye Res 1970: COX in rat: Chen et al. Acta Ophthalmol 1989; COX in mouse: Rueda et al., Mol Vis 2016). The latter paper in adult mouse was not cited in the subsection “Functional mitochondria in photoreceptors”, although quantitative layer-by-layer COX activity was measured and presented. The authors stated that SDH activity was not different between inner and outer retina, but this does not agree with prior published results.*

We are citing these references. Though these assays are fairly common ways to probe mitochondrial activity, our intention was to repeat them ourselves from the standpoint of Ldha enrichment in the photoreceptors. Regarding the reviewer’s second comment, we would like to clarify that we have *not* stated that SDH activity did not differ between inner and outer retina. The statement on line 190-191 (of originally submitted manuscript) is “SDH activity was not lower in photoreceptors relative to INL cells.”

*Seventh, the authors state that "PKLR transcripts were not detected in retina (data not shown)". However, they were found in the retina by other investigators (see Figure 1 of Rueda et al. Mol Vis 2016). No comment is made to acknowledge the difference.*

As mentioned before, we had not come across Rueda et al.’s work while our manuscript was being prepared. From the revision of this manuscript point-of-view, we could not find any statement by the authors on further validation of their microarray results by other means or the changes in their hypotheses if PKLR is found in photoreceptors. This is especially important because James Hurley and colleagues (Lindsay et al. 2014) state that they did not detect PKLR protein in the retina by western blots. It would have been worthwhile to localize these isoforms of pyruvate kinase by IHC and put in perspective of glycolytic propensities. Thus, we cannot comment on their results in efforts to avoid direct contradiction. We are adding a figure supplement to Figure 3 to show our data. The figure supplement information for this figure as well as others that are affected, have been amended. Relevant changes in the text have been made with updated figure information.

*Eighth, the authors should have used a higher resolution microscopy to show specifically the changes observed in the outer segments and possible exclude or determine the possible changes in inner segments and inner segment-mitochondria due to loss of aerobic glycolysis.*

We appreciate the reviewer’s suggestions. While the emphasis of this manuscript was characterization of molecular propensities of the photoreceptors, we are undertaking studies to understand the cell biological implications of aerobic glycolysis in greater detail.

*Ninth, please clarify in the first paragraph of the subsection “Allosteric regulation of glycolysis in photoreceptors”, the three listed criteria points.*

The criteria were conceived as a guide to design an experimental strategy for dissecting glycolytic reliance from housekeeping glycolysis. Since most cells will negatively respond when the core glycolytic pathway is inhibited, we posited that interfering with this pathway was not the correct way of uncovering glycolytic reliance. The points are elaborated below:

1) Does not ablate core glycolytic enzymes in order to avoid pleiotropic effects due to their non-glycolytic roles: It is well known that many glycolytic enzymes (such as PFK1, HK, GAPDH, Enolase) have other roles besides glycolysis. The other enzymes in the pathway might also have roles outside glycolysis that have not yet been appreciated. A classical approach using loss-of-function analysis of glycolytic genes thus has obvious pitfalls, i.e. whether the observed phenotype is really due to their glycolytic role? We wanted to avoid that.

2) Targets glycolytic node such that impact on other biosynthetic pathways such as Pentose Phosphate Pathway (PPP) would be minimal: This was an important criteria for us. PPP has important roles in lipid biosynthesis, NADPH metabolism, nucleotide metabolism and all of these have a significant impact on the biosynthetic makeup of the cell. One could argue that phenotypes arising due to ablation of an enzyme, for example hexokinase, are not necessarily due to impact on glycolysis. Attenuating a cell’s ability to phosphorylate glucose and make it available to other pathways such as PPP can also prove catastrophic. This rationale also holds true for other enzymes such as triose phosphate isomerase (TPI) i.e. does a negative impact due to TPI ablation mean importance of glycolysis or generation of methylglyoxal, a toxic molecule?

3) Uncovers glycolytic reliance and differentiates it from housekeeping glycolysis: Many studies have employed inhibition of glycolytic enzyme activities such as those of GAPDH. The relative usefulness of such an approach should be questioned and care should be taken to not attribute/extrapolate the effects of such a perturbation to aerobic glycolysis. Inhibiting housekeeping glycolysis (by inhibiting glycolytic enzymes that carry reactions from Glucose-6-P to Pyruvate) can have a detrimental effect on many cell types, not only those that carry out aerobic glycolysis.

In this regard our experiments with TIGAR satisfied these criteria. Tigar affects the allosteric regulator of a glycolytic enzyme and thus does not inhibit glycolysis. Very early on in our studies, we deemed the PFK1 node of an interest to us, because slowing down flux through this step would impact commitment of carbons for glycolysis but at the same time can potentially increase PPP, as Karen Vousden’s work has demonstrated. Since fructose-2,6-bisphosphate acts as an allosteric regulator of PFK1, targeting this metabolite satisfied the above criteria.

*Tenth, Ross et al. 2010a should be 2010 as there is no "b".*

We appreciate the careful review. We have now corrected the reference.